# Principles of mitoribosomal small subunit assembly in eukaryotes

Nathan J. Harper[1,2,3], Chloe Burnside[1,2,3] & Sebastian Klinge[1✉]

Mitochondrial ribosomes (mitoribosomes) synthesize proteins encoded within the mitochondrial genome that are assembled into oxidative phosphorylation complexes. Thus, mitoribosome biogenesis is essential for ATP production and cellular metabolism[1]. Here we used cryo-electron microscopy to determine nine structures of native yeast and human mitoribosomal small subunit assembly intermediates, illuminating the mechanistic basis for how GTPases are used to control early steps of decoding centre formation, how initial rRNA folding and processing events are mediated, and how mitoribosomal proteins have active roles during assembly. Furthermore, this series of intermediates from two species with divergent mitoribosomal architecture uncovers both conserved principles and species-specific adaptations that govern the maturation of mitoribosomal small subunits in eukaryotes. By revealing the dynamic interplay between assembly factors, mitoribosomal proteins and rRNA that are required to generate functional subunits, our structural analysis provides a vignette for how molecular complexity and diversity can evolve in large ribonucleoprotein assemblies.

The mitoribosome, which is present in the mitochondria of eukaryotic cells, translates mitochondrial mRNAs that predominantly encode components of oxidative phosphorylation complexes. This molecular machine comprises rRNAs (12S and 16S in humans, and 15S and 21S in yeast) encoded in the mitochondrial genome that associate with predominantly nuclear-encoded mitochondrial ribosomal proteins (mito r-proteins) in the mitochondrial matrix to form the mitoribosome (55S in human and 74S in yeast). Biogenesis of the mitoribosome requires *trans*-acting assembly factors that chaperone rRNA folding and guide mito r-protein incorporation[2–6]. The central role of mitoribosomes to cellular metabolism is highlighted by several human diseases caused by mutations in either mito r-proteins or assembly factors[6–10]. A structural understanding of the principles of mitoribosome assembly is therefore crucial to define the molecular underpinnings of these processes and their associated human diseases.

Although mitochondrial large subunit assembly has been extensively studied in kinetoplastids[7,8] and human cells[9–14], our structural understanding of mitochondrial small subunit (mtSSU) assembly is so far limited to *Trypanosoma brucei*[15,16], an extreme example in terms of evolutionary divergence, and late stages of mammalian mitoribosome assembly[17]. We currently lack fundamental insights into the initial steps of mtSSU assembly in key organisms such as *Saccharomyces cerevisiae* and *Homo sapiens*.

Despite substantial structural divergence across ribosomes from various species[2–6,18–21], all SSU assembly systems must overcome several common rRNA folding hurdles to produce functional subunits. These include the formation of the decoding centre, the key functional module of the SSU, which must be formed in a regulated manner to generate subunits exhibiting high translational fidelity. In addition, the 5′ and 3′ ends of the rRNA need to be generated and integrated into the maturing particle. Simultaneously, premature engagement of SSU assembly intermediates with mature large ribosomal subunits must be prevented to ensure a functional ribosome pool. Although different solutions have evolved for these hurdles in bacterial and eukaryotic systems[22,23], little is known about how yeast and human cells achieve control over these initial assembly steps and how these systems have evolved to catalyse the formation of functional mitoribosomal SSUs that are structurally diverse. To address this, we solved high-resolution structures of six native assembly intermediates from human cells and three native assembly intermediates from yeast cells by cryo-electron microscopy (cryo-EM), revealing several fundamental principles of mtSSU assembly. First, stepwise activities of molecular switches control early rRNA folding events that lay a foundation for the formation of a functional decoding centre. Second, integration of the rRNA 3′ end facilitates compaction of the functional rRNA core, with yeast and human systems each having evolved distinct solutions to do so. Third, recognition and processing of 5′ pre-rRNA is a unique feature of the yeast mtSSU assembly pathway, and is carried out by distinct assembly factors. Fourth, a conserved assembly factor orchestrates maturation of the head domain and prevents premature engagement of mRNA and the mitochondrial large subunit. Together, these principles shed light on the evolution of mtSSU assembly pathways and how species-specific adaptations are used to generate both molecular complexity and diversity.

## Cryo-EM snapshots of mtSSU maturation

To capture human mtSSU assembly intermediates, we used an established CRISPR-based biallelic tagging method to generate a stable HEK293F cell line endogenously expressing the poorly understood

[1]Laboratory of Protein and Nucleic Acid Chemistry, The Rockefeller University, New York, NY, USA. [2]Tri-Institutional Training Program in Chemical Biology, The Rockefeller University, New York, NY, USA. [3]These authors contributed equally: Nathan J. Harper, Chloe Burnside. ✉e-mail: klinge@rockefeller.edu

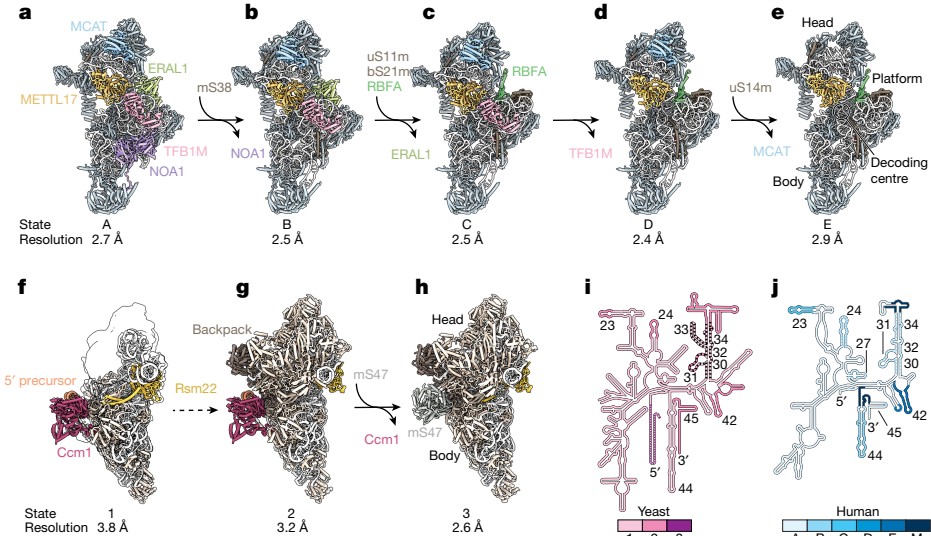

**Fig. 1 | Chronology of mitoribosomal SSU assembly intermediates captured by cryo-EM. a–e**, Atomic models of five human mtSSU assembly intermediates along the assembly pathway. Assembly factors are coloured and changes in protein composition between states are noted. The overall resolution of each cryo-EM reconstruction is noted below each state (see also Supplementary Figs. 2–18). **f–h**, Atomic models of three yeast mtSSU assembly intermediates along the assembly pathway. Assembly factors are coloured and changes in protein composition between states are noted. For state 1 (**f**), the model is shown with the corresponding cryo-EM map filtered to local resolution. The overall resolution of each cryo-EM reconstruction is noted below each state (see also Supplementary Figs. 20–25). The dashed arrow between **f** and **g** indicates multiple maturation events. **i,j**, rRNA secondary structure diagrams of the yeast (**i**) and human (**j**) mtSSU. rRNA elements, which become ordered during the assembly pathway, are coloured according to the state in which the segment becomes ordered, with M corresponding to the mature structure (Protein Data Bank (PDB): 3J9M). Remodelled rRNA is indicated by hashes and processed rRNA is indicated by dots.

assembly factor METTL17 C-terminally tagged with green fluorescent protein (GFP), and confirmed that tagging does not affect mitochondrial translation[24] (Supplementary Fig. 2). Native complexes were isolated by affinity purification, and liquid chromatography with tandem mass spectrometry (LC–MS/MS) revealed the presence of co-purified mtSSU factors and mito r-proteins (Supplementary Data 1). Using cryo-EM, we identified six discrete assembly states (A, B, C, D, E and C*) of the human mtSSU through global and focused classification strategies and solved their structures to resolutions ranging from 2.4 to 3 Å (Fig. 1a–e and Supplementary Figs. 3–19). These intermediates can be ordered into an assembly pathway that rationalizes the maturation status of rRNA elements as well as the presence of assembly factors and mito r-proteins. Although our assignment of states A–E suggests a linear assembly pathway of intermediates bound by METTL17, we cannot exclude the possibility of alternative pathways present in the cell, including those in which certain proteins undergo multiple cycles of binding and dissociation. In particular, the identification of the less populated assembly state C* suggests uncoupling of head and body assembly mechanisms during maturation, supporting the notion that mtSSU assembly does not occur in a strictly linear manner, in line with the parallel assembly pathways previously characterized for bacterial SSU assembly[25] (Supplementary Figs. 9 and 18). These reconstructions reveal the structures and functions of six assembly factors in the context of assembly: the putative methyltransferase METTL17, the GTPases NOA1 (refs. [26,27]) and ERAL1 (refs. [28,29]), the methyltransferase TFB1M[30], the rRNA chaperone RBFA[31], and MCAT (malonyl-CoA-acyl carrier protein transacylase), a protein so far only implicated in mitochondrial fatty acid metabolism[32].

To gain insight into mtSSU biogenesis in yeast, we purified assembly intermediates using two baits (Supplementary Fig. 20). First, by using the yeast-specific factor Ccm1 (ref. [33]) as bait for affinity purification, we obtained two early assembly intermediates, states 1 and 2, which contain both Ccm1 and the METTL17 homologue Rsm22 (Fig. 1f,g and Supplementary Figs. 21,22 and 24–26). State 1 represents the earliest intermediate missing significant modules of mito r-proteins and

exhibits high levels of conformational flexibility in the head region. Second, we purified particles by tagging Rsm22 and solved the structure of a Rsm22-bound intermediate, referred to as state 3 (Fig. 1h and Supplementary Figs. 23–26). LC–MS/MS analysis of both yeast complexes further revealed the presence of co-purified mtSSU assembly factors Mtg3 (homologous to human NOA1) and Rmd9, a protein so far only implicated in the protection of 3′ ends of mitochondrial mRNAs[34,35] (Supplementary Data 2 and 3 and Supplementary Fig. 20). The presence of all four assembly factors (Mtg3, Ccm1, Rmd9 and Rsm22) in an additional assembly intermediate was also confirmed by LC–MS/MS when Mtg3 was used as bait for affinity purification (Supplementary Data 4). However, these particles were not amenable to structural characterization, presumably owing to increased flexibility.

Using our structures of yeast and human mtSSU assembly intermediates, we can rationalize differences in rRNA and mito r-protein composition between yeast and human mtSSUs in the context of assembly, showing how differences are mirrored with contrasting local assembly mechanisms (Extended Data Fig. 1). However, despite the significant divergence in mitochondrial rRNA in these organisms, our reconstructions highlight a shared chronology in which key rRNA segments are remodelled or become ordered as a function of different assembly stages (Fig. 1i,j).

## Guided formation of the decoding centre

The decoding centre is the key functional region of the SSU, which probes mRNA–tRNA interactions to ensure translational fidelity. The controlled formation of the decoding centre is therefore critical for the assembly of functional SSUs. The human assembly system uses two GTPases, NOA1 and ERAL1, which act as molecular switches that promote the first steps in the formation of the decoding centre and the platform (Fig. 2). In state A, NOA1 regulates docking of helix 44 (h44) through two mechanisms. First, an N-terminal extension of NOA1 blocks access to the h44-binding site at the tail of the mtSSU, which is occupied by h44 in state B (Fig. 2e,f). Second, the globular domains of NOA1 hinder h44 docking at the subunit interface while also sequestering

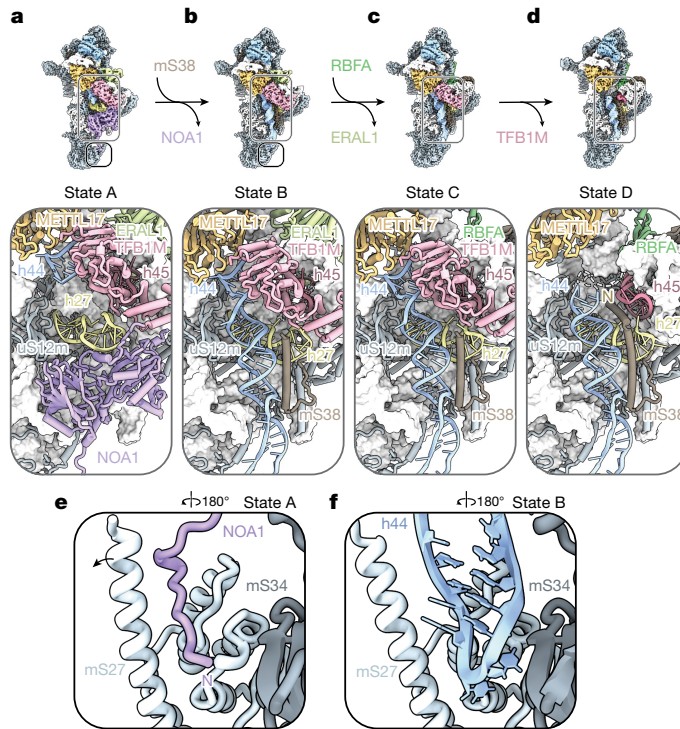

**Fig. 2 | Initial steps of formation of the decoding centre in the human mtSSU. a–d**, Cryo-EM maps of human states A–D (top), with models highlighting stepwise maturation of h44 and the decoding centre (bottom). **e**, Zoomed-in view of the tail region in state A, showing how the N-terminal tail of NOA1 prevents h44 docking. **f**, Zoomed-in view of the tail region in state B, showing accommodation of h44 after NOA1 dissociation.

h27 away from its mature position (Fig. 2a and Extended Data Fig. 2a,b). Dissociation of NOA1 allows for docking of h44 and its stabilization through binding of a helical segment of mS38 as well as repositioning of h27 to form the base of the platform (Fig. 2b).

During eukaryotic and bacterial ribosome assembly, GTPases frequently use conformational changes coupled to nucleotide hydrolysis to promote directional assembly of rRNA and protein modules[36,37]. Within NOA1, we found a vacant GTP-binding site that is partially occluded by a helical insertion of NOA1, which prevents the switch I motif from occupying a conformation competent for GTP binding (Extended Data Fig. 2d). With binding of NOA1 to RNA[26] and stimulation of NOA1-mediated GTP hydrolysis by RNA having previously been shown[27], our structure of NOA1 in physiological context with a maturing mtSSU provides insight into how this subfamily of GTPases, which include the bacterial ribosome assembly factor YqeH, couple a GTPase cycle to rRNA conformational changes in mitochondrial and bacterial SSU assembly[37,38].

Positioned next to NOA1 is the methyltransferase TFB1M, which has a bacterial homologue (KsgA), that methylates adjacent adenosine residues in h45 of the 16S rRNA[39]. In addition to its role in rRNA methylation, discussed in detail in the next section, TFB1M fulfils a second role during mtSSU assembly as one-half of a molecular clamp with METTL17, which stabilizes a segment of h44 that interacts with mRNA during translation[40] (Extended Data Fig. 3a,b). Furthermore, TFB1M initially sequesters h45, preventing h45 from interacting with and stabilizing the mature conformation of h44 until TFB1M methylates two residues in h45 and departs during the transition from state C to state D (Fig. 2a–d and Extended Data Fig. 3c–e). As a result, this molecular clamp inhibits the premature folding of decoding centre rRNA elements and forms an assembly checkpoint in which h45 methylation is required for maturation of the decoding centre. Dissociation of NOA1 and TFB1M during assembly progressively generates the binding site for mS38, which in state D locks the relative orientation of h44, h27 and h45 in a near-mature conformation (Fig. 2a–d). Furthermore, in states C and D,

the rRNA linkers connecting h44 to h45 and h28, which also have direct roles in mRNA decoding during translation, remain flexible due to the presence of the 3′ rRNA-binding assembly factor RBFA, as observed in bacteria[23]. Thus, the functional interplay between the GTPase NOA1, the methyltransferase TFB1M and the RNA chaperone RBFA guide and control the formation of the decoding centre such that h44 docking, stabilization and folding of rRNA linkers occur in a highly controlled and stepwise manner.

The final steps in the formation of the decoding centre require positioning of h31 proximal to the tRNA-binding P site. In human states E and C*, access of h31 to the decoding centre is blocked by assembly factor METTL17, a mechanism that is also present in the yeast system, with the METTL17 homologue Rsm22 performing the same function (Extended Data Fig. 4a–f). Although human mS38 binding occurs in a progressive manner and is present in states B–E, yeast mS38 is not yet bound in state 3, suggesting that mS38 binding and final rearrangements to the decoding site are initiated upon dissociation of Rsm22 (Extended Data Fig. 4g–j).

## Platform and 3′ end rRNA stabilization

The correct formation of the platform and integration of the 3′ end of the mtSSU rRNA is critical for the stability of the decoding centre and hence crucial for effective mitochondrial translation. The general architecture of the platform region in the human mtSSU is similar to its cytoplasmic and bacterial counterparts, with the 3′ rRNA end binding in the back of the platform (Fig. 3a,b). In the maturing human mtSSU, the 3′ end is stabilized in a sequential, mutually exclusive manner by the two assembly factors ERAL1 and RBFA, both of which are conserved between human mitochondria and bacteria. The discovery of ERAL1 within states A and B enables a direct comparison with bacterial systems in which available structural and biochemical data indicate that the binding modes between mitochondrial ERAL1 and bacterial Era are similar[41]. Biochemical studies of bacterial Era further suggest that this

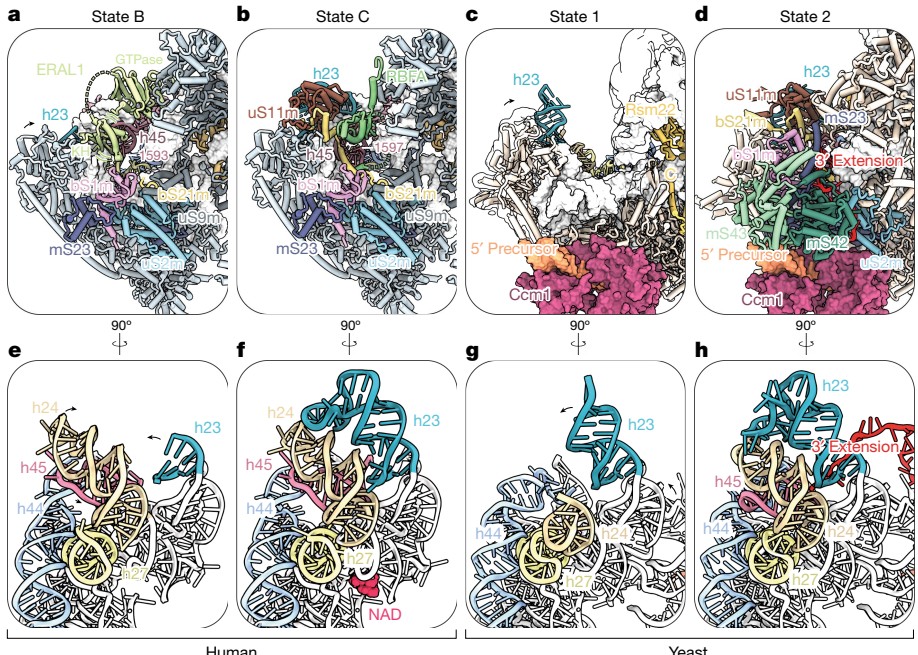

**Fig. 3 | rRNA stabilization of the 3′ end and platform formation are coupled in yeast and human assembly. a**, Model of the architecture around the rRNA 3′ end in human state B, illustrating binding of the 3′ end by ERAL1 and pre-recruitment of bS21m. The arrows highlight platform compaction that occurs between states B and C. **b**, Model of the architecture around the rRNA 3′ end in human state C, showing repositioning of the 3′ end via RBFA binding and recruitment of a uS11m–h23 module through bS21m. **c**, Model of the platform architecture in yeast state 1, before backpack and 3′ rRNA incorporation via the backpack. rRNA h45 is not yet incorporated into the particle, and the arrow highlights platform compaction that occurs between states 1 and 2. **d**, Model of the platform architecture in yeast state 2, after backpack docking and stabilization of 3′ rRNA. In parts a–d, conserved protein and rRNA elements share the same colours. **e**, Side view of the platform region rRNA in human state B. The arrows show movement of rRNA elements occurring between states B and C. **f**, Side view of the platform region rRNA in human state C, revealing compaction of the region, h23 binding and accommodation of NAD. **g**, Side view of the platform region rRNA in yeast state 1. The arrows show movement of rRNA elements occurring between states 1 and 2. **h**, Side view of the platform region rRNA in yeast state 2, revealing compaction of the region and ordering of rRNA h23, h24 and h45.

enzyme binds to the pre-ribosome in a GTP-bound state[39,40], a principle that is probably shared with the mitochondrial assembly system.

In states A and B, ERAL1 binds to the rRNA 3′ end with a C-terminal KH domain in a flexible manner, with the GTPase domain binding uS7m. In state B, the presence of ERAL1 precludes docking of h23 and the complete installation of mito r-proteins bS21m and uS11m. As a C-terminal tethering peptide of bS21m is already observed in state B, it is likely that this segment of bS21m is responsible for the initial docking and eventual incorporation of the bS21m–S11m–h23 module in state C upon dissociation of ERAL1 (Fig. 3a,b). A single point mutation in the human ERAL1 G4 motif results in Perrault syndrome, which is characterized by sensorineural deafness and ovarian dysfunction, suggesting that this mutation compromises the ERAL1 GTPase cycle and hence mtSSU platform assembly[29] (Extended Data Fig. 5a–c).

Although ERAL1 binds in a manner allowing for conformational freedom of the rRNA 3′ end, RBFA binds in a stabilizing manner, sandwiching the free rRNA 3′ end between its KH domain, an N-terminal helical peptide of RBFA, and bS21m (Fig. 3a,b). This composite binding site for the rRNA 3′ end observed here is distinct from that observed in *Escherichia coli*, which is formed exclusively by RBFA[23] (Extended Data Fig. 6e). The bS21m-mediated rigidification of the platform observed between states B and C is accompanied by a concerted rRNA compaction that results in the formation of a NAD-binding site at the base of the platform, which stabilizes a series of single-stranded rRNA linkers in the platform (Fig. 3e,f and Extended Data Fig. 7). Together, the replacement of ERAL1 with RBFA, the docking of h23, uS11m and bS21m, and NAD binding aid in the orchestrated compaction towards a rigid platform.

The gradual compaction of the platform has functional consequences for key enzymatic activities. In bacteria, the TFB1M homologue KsgA mediates dimethylation of h45 adenosine residues, an

activity that has been shown to be promoted by RBFA[31]. Although we did not observe a bound methyl donor in the enzyme, we did observe methylated target residues in states following dissociation of TFB1M (states D and E) (Extended Data Fig. 3e). Our reconstructions provide a structural rationale for this biochemical data, suggesting that TFB1M activity is allosterically regulated by the stability of the platform, which is controlled by the 3′ rRNA-binding assembly factors ERAL1 and RBFA. This mechanism is coupled to upstream events in maturation as a NOA1-bound h27 would preclude platform compaction, ensuring that h44 docking occurs before platform formation and h45 methylation. Maturation events downstream of state E subsequently link RBFA and late-stage assembly factor METTL15 function during mitoribosome assembly with mtIF3-mediated translation initiation[17]. Thus, our data describe how NOA1, ERAL1, RBFA and TBF1M act in concert to enforce and coordinate discrete steps in assembly and provide a unified mechanism for platform formation and methylation licensing. Given that all of these factors are shared between human mitochondria and bacteria, this mechanism is probably shared between these assembly pathways.

Key differences in rRNA 3′ end architecture between the human and yeast mtSSU are reflected in the mechanism of 3′ rRNA stabilization during assembly. The yeast mtSSU contains an extended 3′ rRNA, which reaches into the back of the subunit and is bound by a set of yeast-specific mito r-proteins (mS42 and mS43) in a module that we term the backpack (Fig. 3d and Extended Data Fig. 1g). Our biochemical data indicating that the RNA-binding protein Rmd9 also functions as a mitoribosome assembly factor are consistent with the observation that its depletion destabilizes the 15S rRNA[42] (Supplementary Data 2–4). The 3′ rRNA extension in yeast contains a sequence (5′-AAUAUUCUU-3′) that partially matches a motif that is sequence specifically recognized by Rmd9 (ref.[35]). These data suggest that Rmd9 may be involved in yeast 15S rRNA 3′ end

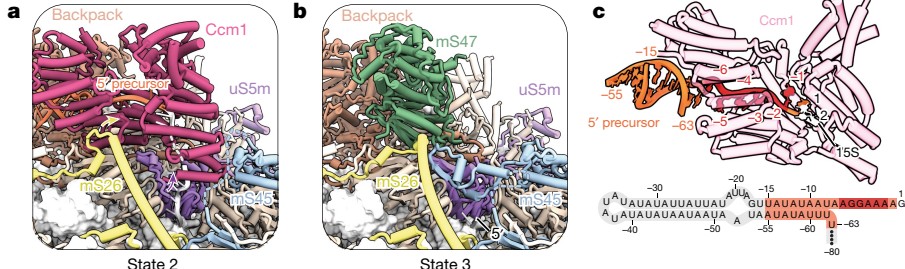

**Fig. 4 | Ccm1 regulates 5′ 15S rRNA maturation in yeast. a**, Model of Ccm1 and the 5′ precursor in state 2. The arrows indicate conformational changes of mito r-proteins during the transition from state 2 to state 3. **b**, Model of the same region in **a**, but in state 3. **c**, Model of Ccm1 bound to the 5′ precursor rRNA (top) and a secondary structure cartoon of the 5′ precursor sequence (bottom), coloured as in the model: the red region is specifically recognized by Ccm1, the orange regions are resolved, and the grey regions are not resolved.

maturation in an early and likely conformationally dynamic assembly intermediate. The presence of an extended 3′ rRNA, mS42 and mS43, and the absence of RBFA or ERAL1 in *S. cerevisiae* suggests that stabilization of the 3′ end occurs in a distinct way compared with human and bacterial systems. Consistent with this idea, this backpack module and h45 are flexible and unresolved in state 1, whereas in state 2, these elements have been docked onto the particle (Fig. 3c,d). Docking of the backpack module and 3′ rRNA stabilization results in platform compaction and rRNA ordering similar to that observed in humans (Fig. 3g,h). Given the unique way in which yeast mito r-proteins mS42 and mS43 stabilize the 15S rRNA 3′ end from state 2 onwards, it appears that these mito r-proteins have functionally replaced human assembly factors ERAL1 and RBFA. Instead of the controlled 3′ rRNA stabilization mediated by human ERAL1 and RBFA, which is coupled to TFB1M activity, yeast mito r-proteins mS42 and mS43 mediate 3′ rRNA docking and stabilization while remaining bound in the mature mtSSU. Together, these data illustrate the link between 3′ end stabilization and platform maturation and show how rRNA and mito r-protein expansion that occurred through evolution in yeast are accommodated in the yeast assembly system.

## Ccm1 controls 15S rRNA 5′ end processing

Unlike human mitoribosomal RNAs, which are punctuated with tRNAs that are post-transcriptionally cleaved and processed to release 'full-length' rRNAs, yeast rRNAs must be processed by exonucleases to generate mature subunits[43]. In states 1 and 2, the pentatricopeptide repeat protein Ccm1 localizes to the 5′ precursor of the 15S rRNA in a position that is subsequently occupied by mS47 both in state 3 and in the mature yeast mtSSU (Fig. 4a,b). The binding site and observed function of Ccm1 explain previous biochemical data that implicate this protein in the stabilization of 15S rRNA[33]. Ccm1 uses structure and sequence-specific RNA recognition, binding to a single-stranded region of the precursor and specifically recognizing bases −6 to −1 (Fig. 4c). Ccm1 is therefore in the unique position to chaperone mitochondrial pre-rRNA in a co-transcriptional manner similarly to eukaryotic ribosome assembly factors binding to other 5′ precursor elements[44]. The exchange of Ccm1 for mS47 is coupled to the irreversible removal of precursor rRNA that is accompanied by conformational changes of the mito r-proteins uS5m and mS26m, and additional ordering of a flexible linker of mS45 in state 3 that is previously absent in state 2 (Fig. 4a,b). These conformational changes signal completion of 5′ end rRNA processing through protection of the mature 5′ end of the 15S rRNA and stabilization of mS47 in state 3. The removal of the 5′ precursor together with the dissociation of Ccm1 is at least in part catalysed by the 5′–3′ exoribonuclease Pet127, which has been previously implicated in 15S rRNA and mitochondrial mRNA processing[45]. Consistent with a role in pre-rRNA processing, we found that the deletion of the gene encoding Pet127 results in the accumulation of the 5′ precursor and also affects the association of both translational activators of a small number of

mitochondrial mRNAs as well as mtSSU assembly factors with mtSSUs and their precursors purified via the tagged early binding mito r-protein uS17m (Supplementary Fig. 27 and Supplementary Data 5). Together, our structural and biochemical data show that Ccm1 and Pet127 coordinate the controlled processing of the 5′ end of yeast mitochondrial rRNA.

## Related mechanisms of head assembly

The head domain of the ribosomal SSU is responsible for ratcheting along mRNA and tRNA binding during translation. Our reconstructions reveal how assembly factors promote stepwise transformations of rRNA and protein elements in the mtSSU head region. A key conserved assembly factor within mitochondria, which uses steric hindrance during the maturation of the head region, is the putative methyltransferase Rsm22 (METTL17 in human). In both yeast and human systems, Rsm22 (METTL17) binds at the interface of the head and body domains, occluding the mRNA channel and preventing compaction of the head domain towards the body (Fig. 5a,e). The presence of this factor across assembly intermediates emphasizes its central role as a gatekeeper against premature mRNA binding in earlier stages of assembly, which is coupled to its function in sequestering h31, one of the last elements of the decoding centre to be positioned (Extended Data Fig. 4c–f). We have further identified an Fe$_4$S$_4$ cluster as an architectural cofactor of the Rsm22 (METTL17) protein family, which rationalizes previous structural data obtained in kinetoplastids, as well as genetic interactions between METTL17 and iron–sulfur cluster biogenesis machinery[15,46] (Extended Data Fig. 8a–c). The binding of METTL17 to h31 also prevents association of the late-stage assembly factor METTL15, which is only observed in assembly intermediates with a mature head domain[17]. These events enforce a chronology of human mtSSU assembly in which head formation and compaction precede final METTL15-mediated modification of the decoding centre. The architectural role of METTL17 and position in the assembly pathway upstream of METTL15 would further explain rRNA methylation defects observed in METTL17-deficient cells[47].

Our structures show that in humans, METTL17 orchestrates maturation of head rRNA through interplay with an additional assembly factor identified in this study: MCAT (Fig. 5a–d). In states A–D, the presence of both of these factors prevents compaction of the 3′ major rRNA domain, splaying rRNA elements and protein modules apart to allow for significant intermodule flexibility. In addition to sterically hindering folding and docking of h42, which forms a lid along the top of the head domain, MCAT sequesters h31, a primarily single-stranded rRNA segment emanating from h30 and h32, from the core of the head domain (Fig. 5a,b). Upon the departure of MCAT in state E, h42 can become ordered in the position previously occupied by MCAT and h31 can adopt a compact conformation forming contacts with h34, h42 and h43 (Fig. 5c). Thus, dissociation of MCAT triggers global compaction of the 3′ major domain. The identification of MCAT as a bona fide mtSSU assembly factor recontextualizes the observation that mutations in

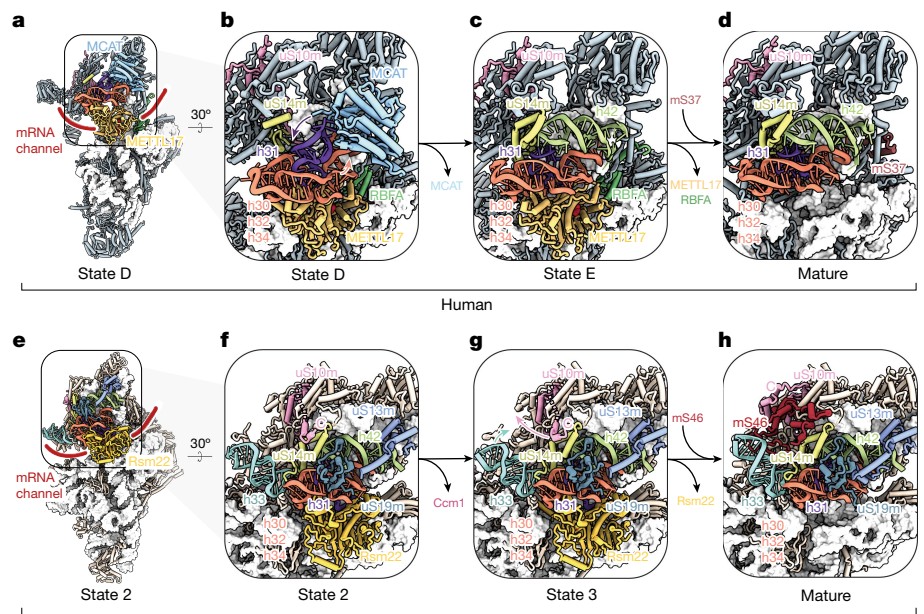

**Fig. 5 | Maturation of the human and yeast mtSSU head domain. a**, Model of human state D, highlighting inhibition of mRNA binding via METTL17. **b–d**, Top-down zoomed-in view of the head domain across human assembly intermediate state D (**b**), state E (**c**) and the mature pre-initiation structure (PDB: 6RW4) (**d**). **e**, Model of yeast state 2, highlighting inhibition of mRNA binding via Rsm22. **f–h**, Top-down zoomed-in view of the head domain across

yeast assembly intermediate state 2 (**f**), state 3 (**g**) and the mature structure (PDB: 5MRC) (**h**). Binding and dissociation of protein factors occurring between states are noted between panels. Across all panels, conserved protein and rRNA elements share the same colours, and coloured arrows show conformational changes in RNA or protein elements that occur during assembly.

this protein causes LHON-like optic neuropathy[32]. Several mutations in mtDNA encoding subunits of respiratory chain complexes are also associated with this disorder, suggesting that functional defects from these MCAT mutations stem from its role as an mtSSU assembly factor rather than its role in fatty acid metabolism (Extended Data Fig. 5d–f).

Although yeast lack an MCAT homologue, the yeast mtSSU has two additional mito r-proteins present in the same location that MCAT binds in humans, suggesting that they may chaperone h42 docking early in assembly. Although Rsm22 also prevents head compaction in yeast, unlike its homologues in human and *T. brucei* systems, yeast Rsm22 has lost SAM-binding capacity but has retained the overall methyltransferase fold (Extended Data Fig. 8a–c). A C-terminal helical peptide segment of uS10m initially prevents the binding of mS46,

the final mito r-protein to bind to the head domain. Dissociation of Rsm22 subsequently drives reorientation of h33 and repositioning of this peptide to allow mS46 binding and complete maturation of the head domain (Fig. 5g,h).

In both yeast and human mtSSU assembly systems, maturation of the head domain involves assembly factors acting to sterically hinder discrete rearrangements in rRNA and conserved ordering of mito r-protein elements to promote rRNA packing (Extended Data Fig. 8d–g). In yeast, protein elements such as uS10m seem to have evolved to have an active role during mtSSU assembly by preventing rRNA rearrangements and mito r-protein association while becoming permanent constituents of the mature subunit. This evolutionary trajectory towards dual roles of mito r-proteins in yeast is akin to the adaptations observed for binding, stabilization and incorporation of the 3′ end of the rRNA, in which assembly factors used in human and bacterial systems are exchanged for mito r-proteins binding to an extended rRNA segment.

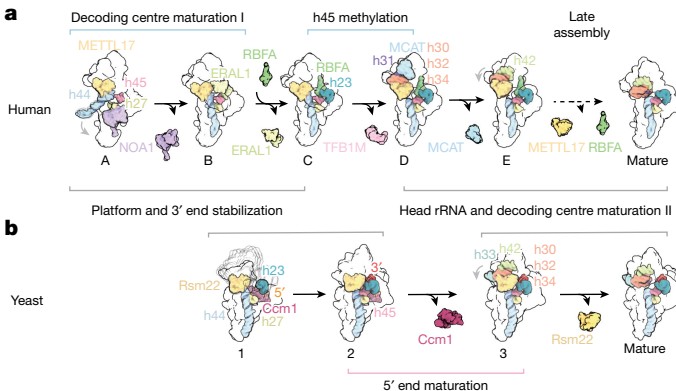

**Fig. 6 | Shared principles of mitoribosomal SSU assembly in yeast and human cells. a,b**, Schematic of human (**a**) and yeast (**b**) mtSSU assembly pathways. The centre brackets indicate common themes across both assembly systems, whereas the outer brackets indicate evolutionarily divergent assembly solutions. Assembly factor binding and dissociation events are shown. Dashed arrows indicate multiple maturation events[17].

## Discussion

A fundamental challenge during the formation of ribosomal SSUs is the correct assembly of functionally critical rRNA regions such as the decoding centre. Doing so in a controlled and regulated manner is essential for the generation of functional subunits that exhibit high translational fidelity to control the quality of newly synthesized proteins. For mitoribosomes, this aspect of assembly regulation is especially crucial given that its translational products, subunits of oxidative phosphorylation complexes, are essential for creating ATP under aerobic conditions. Here we have solved structures of both human and yeast mitoribosomal SSUs undergoing assembly under native conditions. In addition to uncovering key architectural features of these pre-mitoribosomal subunits and illuminating the roles of assembly factors in regulating the formation of rRNA elements, we have shown how assembly mechanisms and machinery have evolved to solve shared hurdles during assembly between these two model organisms (Fig. 6).

In the human assembly system, we have shown that assembly factors in concert with mito r-proteins catalyse stepwise formation and stabilization of rRNA elements before states previously observed[17]. In particular, a network of assembly factors (NOA1, ERAL1, RBFA, TFB1M and METTL17) couple maturation of the decoding centre to platform compaction and 3′ end rRNA stabilization, in the process licensing methylation of key residues in h45 at the correct time during assembly (Fig. 6a). Although the mechanism of 3′ end stabilization in yeast is distinct from that in humans, requiring unique mito r-proteins to bind to and dock an extended rRNA 3′ end rather than dedicated 3′ end binding assembly factors as seen in the human system, we show that stable incorporation of the rRNA 3′ end is a key checkpoint in mtSSU assembly (Fig. 6b). Similarly, yeast uses a distinct 5′ end rRNA maturation pathway involving a unique assembly factor and yeast-specific mito r-proteins. Conversely, the shared architecture and role of Rsm22 (METTL17) in prevention of mRNA binding, subunit compaction and formation of the decoding centre demonstrates a level of conservation between these two assembly systems, a concept additionally highlighted by similarities in rRNA-folding chronology (Figs. 1i,j and 6). We further note that our study provides a structural view of assembly intermediates that can be associated with Rsm22 (METTL17), thereby visualizing one assembly line of what is probably a branched assembly landscape involving assembly factors dissociating and re-associating at key junctions.

The similarities and differences shown here illustrate how different molecular mechanisms involving dedicated assembly factors and bifunctional mito r-proteins have evolved to solve common hurdles during assembly. These principles highlight evolutionary coupling between assembly mechanism and the final architecture of large, multimeric biomolecular complexes, and shed light on the emergence of molecular complexity and diversity across species.

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

## Methods

### HEK293F cell line generation and cell culture

To generate a stable mammalian cell line with endogenously tagged *METTL17* on both alleles, we used a genome editing platform previously developed in the laboratory and used for purification of human pre-ribosomes called SNEAK PEEC (surface engineered fluorescence assisted kit with protein epitope enhanced capture)[24]. This system uses guide RNA-targeted Cas9 genome cleavage and in-frame cell-surface epitope expression to allow for identification of biallelically edited clones by fluorescence-activated cell sorting (FACS). Repair template plasmids contain 600-bp homology arms flanking the Cas9 cut site located in the last exon of *METTL17*, and an in-frame insert containing the affinity tag (linker-twinStrep-FLAG-10xHis-TEV-SPOT-3C-GFP) and one of two engineered cell surface epitopes (the Btuf vitamin $B_{12}$ binding protein from *E. coli* (PDB: 5OVW) and the capsid protein p24 from human immunodeficiency virus 1 (PDB: 5O2U)) separated by a T2A site[48,49].

To generate these repair templates, homology arms flanking the Cas9 target site and a multiple cloning site were cloned into the pUC57 backbone (GenScript). The entire insert sequence, including the affinity tag, T2A site, and the surface display epitopes, were introduced using NotI and PacI sites in the multiple cloning site. The plasmid for expression of single guide RNA and *Streptococcus pyogenes* Cas9 (variant eSpCas9(1.1)) was derived from Addgene plasmid #71814, a gift from F. Zhang[50]. The 20-bp single guide RNA target sequence (5′-GTTTCCTCCATCTACGGCTC-3′, PAM: CGG) was designed using crispor.telfor.net[51] and cloned into this vector using BbsI sites.

HEK293F cells (R79007, Thermo Fisher Scientific) were grown in 24-well plates (353047, Falcon) in Freestyle 293 expression medium (12338026, Thermo Fisher Scientific) supplemented with 2% heat-inactivated fetal bovine serum (FBS). Cell lines were not tested for mycoplasma. Each well was transfected using 1 μg of total DNA (both repair template plasmids and the Cas9/single guide RNA plasmid in equimolar concentrations) at approximately 90% confluency using the transfection reagent Lipofectamine 2000 (11668019, Thermo Fisher Scientific) in 500 μl Opti-MEM medium (31985070, Thermo Fisher Scientific). Twelve hours after transfection, cells were washed and resuspended in Freestyle 293 expression medium supplemented with 2% heat-inactivated FBS. After recovery, cells were expanded to six-well plates (10062-892, VWR) and passaged over the course of 14 days.

To prepare cells for flow cytometry, cells were harvested from six-well plates by gentle aspiration, washed in 1X PBS with 0.1% BSA, and resuspended in 1X PBS with 0.1% BSA to a concentration of $1–10 × 10^6$ cells per millilitre. Next, two fluorescently labelled nanobodies, each of which specifically binds to one of the epitopes expressed at the cell surface, were added to cell suspensions at a concentration of 10 nM. The labelling reaction was carried out in the dark for 30 min on ice before washing cells twice with 1X PBS with 0.1% BSA. Cells were then resuspended in 200 μl of 1X PBS with 0.1% BSA and filtered directly before cell sorting to remove cell clumps (352235, BD Falcon). Cell sorting was carried out using a BD FACSAria cell sorter (BD Biosciences) using FACSDiva Software (BD Biosciences). DAPI was used to stain dead cells directly before sorting. Necessary single-colour compensation controls, including GFP and each individual labelled nanobody, were performed directly before sorting of experimental cells using HEK293F cells expressing GFP or cell-surface epitopes from plasmids. To identify biallelically tagged METTL17 cells, cells were first sorted to identify GFP-positive cells, followed by a second selection to isolate cells staining positive for both btuF and p24 surface epitopes. Data analysis was carried out using FlowJo (FlowJo, LLC).

Single-cell clones were sorted into 96-well plates containing 200 μl Freestyle 293 expression medium (12338026, Thermo Fisher Scientific) supplemented with 2% heat-inactivated FBS per well. After 2 weeks of clonal expansion, individual clones were screened by two PCRs to confirm biallelic editing. Genomic DNA was isolated using QuickExtract DNA Extraction Solution (QE09050, Lucigen) according to the manufacturer's protocol. Of extracted solution, 30 μl was used per PCR (50 μl final reaction volume) using the following PCR primers:

For reaction 1, forward: 5′-TATGTGTTCACAGTGTCCCCATGAACT CCC-3′ (genomic primer annealing upstream of the METTL17 left homology arm); reverse: 5′-GACTCCACGGGGCCAACTGTCTCAAGG-3′ (BtuF epitope specific).

For reaction 2, forward: 5′-TATGTGTTCACAGTGTCCCCATGAACT CCC-3′ (genomic primer annealing upstream of the METTL17 left homology arm); reverse: 5′-TGCTGTCATCATTTCCTCGAGCGTAGCACC-3′ (HIV p24 epitope specific).

Biallelically tagged METTL17 HEK293F cells were subsequently grown in suspension in Freestyle 293 expression medium at 37 °C with 80% humidity at 8% $CO_2$, shaking at 135 r.p.m.

### Yeast strains

*S. cerevisiae* strain BY4741 (MATa his3Δ leu2Δ0 met15Δ0 ura3Δ0) was used as the starting point for construction of endogenously tagged strains. All strains were grown aerobically at 30 °C in YPG media (1% yeast extract, 2% peptone and 3% glycerol). Assembly intermediates of the *S. cerevisiae* mtSSU were purified from three genetically modified BY4741 strains: the first strain containing a 3C cleavable C-terminal GFP-tagged *MTG3* (MATa his3Δ leu2Δ0 met15Δ0 ura3Δ0 *MTG3*-linker-3c-GFP::ClonNat), the second strain containing a TEV protease-cleavable C-terminal mCherry-tagged *CCM1* and a 3C cleavable C-terminal GFP-tagged *MTG3* (MATa his3Δ leu2Δ0 met15Δ0 ura3Δ0 *MTG3*-linker-3c-GFP::ClonNat *CCM1*-linker-tev-mCherry: :HphMX4), and the third strain containing a C-terminal tandem 3C protease-cleavable mCherry and TEV-cleavable alfa peptide tag on *RSM22* (MATa his3Δ leu2Δ0 met15Δ0 ura3Δ0 *RSM22*-linker-tev-alfa tag-3C-mCherry::ClonNat).

For comparison of mitoribosomal complexes in wild-type and *PET127*-knockout (KO) backgrounds, a *PET127*-KO strain was purchased from the Yeast Knockout (YKO) Collection of Horizon Discovery (MATa his3Δ leu2Δ0 met15Δ0 ura3Δ0 *PET127*::kanMX6). Wild-type and *PET127*-KO strains were modified to contain a TEV protease-cleavable C-terminal GFP-tagged *US17M* (MATa his3Δ leu2Δ0 met15Δ0 ura3Δ0 *US17M*-linker-tev-GFP::HphMX4, MATa his3Δ leu2Δ0 met15Δ0 ura3Δ0 *PET127*::kanMX6 *US17M*-linker-tev-GFP::CloNat).

All strains were generated by applying standard genomic tagging techniques. Primers used for yeast genomic tagging are detailed: Ccm1 forward primer: AAAAGTCACACGCAAAGGCCCTGAAGTGG GAGGAACAAGAACTTAACATGACGCTGCAGGTCGACGGATCC; Ccm1 reverse primer: GAGCTTTTATTTACAGGCGCTTTACTATATA TATATATGTGCATGCATGGGGCAGATCCGCGGCCGCATAGG. Mtg3 forward primer: TGGTTCAACACCTAGAAAAAAAAAAGAATCC CTATTGGTACTACCAATGGACGCTGCAGGTCGACGGATCC; Mtg3 reverse primer: TGAATATACTTTGTTCCCAACTTTCTTTTTAGTT TACTTACATTCATATTGGCAGATCCGCGGCCGCATAGG. Rsm22 forward primer: AATTGTCCACTTTTCACGGCAACGATTTTTTA CAACATGTAAATAGAAAAACGCTGCAGGTCGACGGATCC; Rsm22 reverse primer: CAAACTGTACTATATATATATCATATTCACGTATGTA GAATATTAAAGTAGGCAGATCCGCGGCCGCATAGG. uS17m forward primer: AAAATTTGTTAAGAAAGCACGTCATGATGGATATGCAACAAC CAAGCCAGACGCTGCAGGTCGACGGATCC; uS17m reverse primer: TACTATTTACGTAACGAAGTCTTGTAGTTGTATATATTTACAGAGTGTAGG GCAGATCCGCGGCCGCATAGG.

The described strains were grown aerobically at 30 °C in YPG media (1% yeast extract, 2% peptone and 3% glycerol) until an optical density at 600 nm of 1.5–2. Cells were harvested at 4,000$g$ for 5 min, washed once with 1 l of ice cold dd$H_2O$ and once with a volume of dd$H_2O$ supplemented with protease inhibitors (E64, pepstatin and PMSF) equal to the mass of the cell pellet. The final pellet was flash frozen in liquid nitrogen and (if used for complex purification) lysed by cryo-grinding

using a Retsch Planetary Ball Mill PM100. Cryo-ground powder was stored at −80 °C or used immediately.

## Purification of human pre-mitoribosomal complexes

Biallelically tagged METTL17 HEK293F cells were grown in suspension in Freestyle 293 expression medium at 37 °C with 80% humidity and 8% $CO_2$, shaking at 135 r.p.m. Of cells, 3.2 l were grown to $3.0 \times 10^6$ cells per millilitre and harvested by centrifugation at 2,000$g$ (JLA 8.100, Beckman Coulter). Cells were washed in ice-cold 1X PBS, dropped into liquid $N_2$ and frozen at −80 °C until use. Crude mitochondrial purification was completed using a protocol adapted from ref. [52]. All steps were completed on ice using pre-cooled buffers at 4 °C, and all centrifugation steps were completed at 4 °C. Frozen cells were resuspended in ice-cold hypotonic buffer (20 mM HEPES-KOH pH 7.5, 5 mM KCl, 1.5 mM $MgCl_2$ and with 1X protease inhibitors (E64, pepstatin and PMSF)) at a ratio of 1 ml g$^{-1}$ of frozen material. Cells swelled for 10 min before the resuspension was transferred to a dounce homogenizer. Cells were then lysed by 15 strokes of pestle B. 2.5X MSH buffer (525 mM mannitol, 175 mM sucrose, 20 mM HEPES pH 7.5 with 1X protease inhibitors (E64, pepstatin and PMSF)) was added to 1X concentration. Lysate was then cleared by two rounds of centrifugation at 1,300$g$ for 5 min to sediment cell debris. Supernatant was then spun at 15,000$g$ for 15 min (JA 25.50) to pellet mitochondria. Crude mitochondria were washed with 1X MSH buffer and resuspended in mitochondrial resuspension buffer (25 mM HEPES-KOH pH 7.5, 100 mM KCl, 25 mM Mg(OAc)$_2$ and 1% Triton X-100 v/v with 1X protease inhibitors (E64, pepstatin and PMSF)) at a ratio of 1 ml per 2.5 g frozen material. Mitochondria were lysed for 30 min at 4 °C with constant mixing, and lysate was cleared by centrifugation at 17,000$g$ for 20 min (JA 25.50, Beckman Coulter). Of pre-equilibrated anti-GFP beads (Chromotek), 30 µl was added to the supernatant and incubated for 3 h at 4 °C with constant mixing. Beads were washed twice with mitochondrial resuspension buffer, and once with wash buffer (25 mM HEPES-KOH pH 7.5, 100 mM KCl and 25 mM Mg(OAc)$_2$ with 1X protease inhibitors (E64, pepstatin and PMSF)). Beads were then resuspended in 40 µl elution buffer (25 mM HEPES-KOH pH 7.5, 100 mM KCl, 25 mM Mg(OAc)$_2$ and 0.005% C12E8), and eluted by addition of 3C protease over 1 h on ice. Eluted complexes were used immediately for grid preparation, with analysis by LC–MS or SDS–PAGE.

## Cryo-EM grid preparation and data acquisition of human complexes

Grids were prepared using a Vitrobot Mark IV robot (FEI Company) at 95% humidity. Of eluted complex solution, 3.5 µl was applied to a glow discharged Quantifoil Au R3.5/1 with a layer of 2-nm ultrathin carbon (LFH7100AR35, Electron Microscopy Sciences). The grid was incubated for 2.5 min inside the Vitrobot chamber before manually blotting excess solution. Of fresh sample, 3.5 µl was then reapplied before another 2.5-min incubation and manual blotting. A final 3.5-µl sample application and 2.5-min incubation was completed before blotting with a blot force of 8, a blot time of 8 and plunging into liquid ethane. Grids were imaged on a Titan Krios electron microscope (FEI) with an energy filter (slit width of 20 eV) and a K3 Summit detector (Gatan) operating at 300 kV with a nominal magnification of ×64,000. SerialEM[53] was used to collect a dataset totalling 37,047 micrographs with a defocus range of −0.7 to −1.8 µm and a super-resolution pixel size of 0.54 Å. Micrographs contained 42 frames using a total dose of 28.5 e$^-$ per pixel per second with an exposure time of 2.1 s, resulting in a total dose of 51 e$^-$ Å$^{-2}$. Using a multi-shot strategy, nine micrographs were collected per hole, and four holes were imaged at each stage position. A second dataset comprising 9,990 micrographs was collected on the same grid on different holes using a different collection strategy in which nine holes were imaged at each stage position, using the same defocus range, electron dose and frame count. These datasets are denoted dataset 1 (37,047 micrographs) and dataset 2 (9,990 micrographs) in the supplementary figures and below.

## Cryo-EM data processing of human complexes

All cryo-EM processing steps were completed using RELION 3.1.1 (ref. [54]) unless noted, and are summarized in Supplementary Figs. 3–18. For each dataset, images were gain corrected, dose weighted, aligned and binned to a pixel size of 1.08 Å using MotionCor2 implementation in RELION[55], and micrograph defocus was estimated using GCTF1.18 (ref. [56]). An initial model for 3D classification was generated by using a test subset of 2,597 micrographs from dataset 1 as follows: CrYOLO[57] was trained and used to pick 533,056 particles from these micrographs, and these particles were extracted at 8.64 Å per pixel (8× binning). After 2D classification ($K = 100$ and $T = 2$), six classes were conservatively picked, which resembled different views of the mature mtSSU, resulting in 65,174 particles, which were subsequently re-extracted at a pixel size of 4.32 Å per pixel (4× binning). These particles were used to generate an ab initio initial model. This initial model was used as a reference for 3D classification with global alignment of a second set of 333,265 particles from the 2D classification, which were picked less conservatively and re-extracted at 2.16 Å per pixel (2× binning). Two resulting classes, comprising 80% of the data and 268,009 particles, were combined and refined to 4.36 Å. This reconstruction was used as a reference for classification of the entire dataset.

CrYOLO was trained independently on each dataset and used to pick particles from all micrographs, resulting in 7,597,700 particles for dataset 1 and 1,511,635 particles for dataset 2. These particles were extracted at 8.64 Å per pixel (8× binning), and dataset 1 was split into four subsets with 1,894,925 particles each. To clean the particle stack, each subset, as well as all particles from dataset 2, were subjected to 2D classification ($K = 100$ and $T = 2$) followed by 3D classification with alignment ($K = 5$ and $T = 4$; 25 iterations with 7.5° angular sampling, 10 iterations with 3.7° angular sampling and 10 iterations with 1.8° angular sampling). Good classes from each 3D classification were combined and duplicates were removed, resulting in 1,927,789 total particles, with particles from each dataset having different optics groups. Contrary to the test dataset, the selected particles comprised approximately only 40% of particles from each subset. To ensure all good particles were selected, all particles from initially unselected classes were combined and re-extracted at 4.32 Å per pixel (4× binning), totalling 4,015,406 particles. These particles were split into two subsets of 2,007,700 particles each and subjected to a second round of 3D classification with alignment ($K = 5$ and $T = 4$; 25 iterations with 7.5° angular sampling, 10 iterations with 3.7° angular sampling and 10 iterations with 1.8° angular sampling). Good classes were combined and duplicates were removed, resulting in 1,183,364 particles, which were merged with the initial selection of 1,927,789 particles. After removing duplicates, a final particle stack with 2,575,411 particles was re-extracted at 1.08 Å per pixel (1× binning), refined and subjected to Bayesian polishing and contrast transfer function (CTF) refinement. A subsequent refinement resulted in a 2.45 Å consensus map.

To determine the compositional and conformational heterogeneity present in the consensus map, 3D classification without alignment was used. Initial attempts were done without a mask, which resulted in classes with different relative orientations between the head and body domains, reflecting the conformational freedom of the head domain. However, this strategy was not optimal for distinguishing compositional differences in the data, especially those present in the body, platform and head regions. As a result, 3D classification without alignment ($K = 10$ and $T = 30$) was performed with a mask around the body domain, which successfully resolved the compositional heterogeneity present. Four classes were identified, which represented compositionally unique assembly intermediates. Of the data, 10% (263,184 particles) represent the earliest state in which h44 is displaced, the density for NOA1 and TFB1M are present and the platform is less ordered. We denote this intermediate state A. Of the data, 19% (497,031 particles) represent the next state, in which h44 is present and NOA1 is absent, but the platform

is still remodelled and TFB1M is present, as in state A. We denote this intermediate state B. Of the data, 10% (262,363 particles) represent a state in which the platform is well ordered and TFB1M is present. We denote this intermediate state C. Of the data, 27% (697,003 particles) represent a state in which TFB1M is not present and the platform is well ordered. We denote this intermediate state D. Each of these states were subjected to another round of CTF refinement and Bayesian polishing, refined and subsequently processed independently.

To fully structurally characterize these intermediates, we adopted a strategy using focused classification and/or focused refinement to improve local resolutions of flexible regions, such as those occupied by assembly factors, the head and the platform. In particular, the head domain exhibits significant conformational freedom relative to the body domain, as is seen in other ribosomal SSU reconstructions. For each state, focused refinement was used to realign the particles on the head domain, which revealed significant remodelling of rRNA due to the presence of the assembly factors METTL17 and MCAT. The resulting modules exhibit additional relative flexibility, further limiting resolution of these regions. To address this, the head domain for each state was subjected to focused 3D classification without alignment to identify a subset of particles with well-defined modules before focused classification and/or focused refinement of individual modules. For states C and D, this classification resulted in the identification of small subsets of data in which MCAT is absent and h42 is stabilized in a near-mature conformation (Supplementary Figs. 6c and 7d). We denote these states state C* and state E, respectively, and further processed these particles independently (Supplementary Figs. 8 and 9). We propose that state C* represents an intermediate of a secondary pathway in which MCAT dissociation occurs before dissociation of TFB1M (thus skipping state D), whereas state E represents an intermediate directly following state D and state C*. All focused refinements were performed using solvent flattening, and reported resolutions are based on the gold-standard FSC (Fourier Shell Correlation)-0.143 criterion as calculated during post-processing in RELION. Local resolutions of focused maps were calculated in RELION.

To provide a complete view of the assembly process, we exploited overlap in body and head maps to generate composite maps and corresponding models that represent the predominant relative conformation between these domains. Focused maps were fit and resampled into the overall maps in Chimera[58] and were subsequently combined to create composite maps using phenix.combine_focused_maps[59]. Half-maps from each focused refinement were also combined using the same procedure. The composites were then rescaled to a calibrated pixel size of 1.062 Å per pixel, and overall resolutions were calculated in RELION. cryoSPARC[60] was used to calculate local resolution of composite maps and to generate maps filtered by local resolution. The Remote 3D-FSC Processing Server was used to calculate 3D-FSC curves for composite maps[61].

### Model building and refinement of human complexes

For human states, a starting model including all ribosomal proteins and 12S rRNA (PDB: 6RW4) was used as an initial template for model building by rigid body docking into composite maps. To build assembly factors, a combination of homology models generated in the Alpha-Fold database (NOA1, METTL17, RBFA and ERAL1) and existing X-ray structures (MCAT (PDB: 2C2N) and TFB1M (PDB: 6AAX)) were used as starting models before manual adjustment using COOT[62]. Changes in RNA conformation were accounted for in each state by manual refinement in COOT[62]. Finally, entire models for each state were adjusted with three cycles of refinement in PHENIX using phenix.real_space_refine using secondary structure restraints[59]. Owing to lower local resolution for the ERAL1 KH domain, we trimmed side chains in this section of the chain to $C_\beta$ after all-atom refinement. Model refinement statistics for human models can be found in Extended Data Table 1.

Maps and models were visualized in Chimera[58], ChimeraX[63] and PyMol (Schrödinger, LLC).

Figures were generated using ChimeraX[63]. PyMol sessions are available for each state individually and with all states together.

### Purification of Ccm1-associated mtSSU intermediates

Cryo-ground yeast powder was resuspended in buffer A (25 mM HEPES-KOH pH 7.5, 100 mM KCl, 25 mM MgOAc, 0.1% Triton X-100, 2 mM DTT, PMSF, pepstatin and E64) and cleared by centrifugation at 4 °C and 40,000g for 20 min. The cleared lysate was incubated with NHS–sepharose beads (Cytiva) coupled to anti-mCherry nanobodies for 3 h at 4 °C on a nutator. Beads were pelleted by centrifugation at 4 °C for 1 min at 127g. After three washes in buffer A, complexes were eluted in buffer B (25 mM HEPES-KOH pH 7.5, 100 mM KCl and 25 mM MgOAc) supplemented with TEV protease for 1 h at 4 °C.

### Mtg3-associated mtSSU intermediate purification

Cryo-ground yeast powder was resuspended in buffer A (25 mM HEPES-KOH pH 7.5, 100 mM KCl, 25 mM MgOAc, 0.1% Triton X-100, 2 mM DTT, PMSF, pepstatin and E64) and cleared by centrifugation at 4 °C and 40,000g for 20 min. The cleared lysate was incubated with anti-GFP nanobody beads (Chromotek) for 3 h at 4 °C on a nutator. Beads were pelleted by centrifugation at 4 °C for 1 min at 127g. After three washes in buffer A, complexes were eluted in buffer B (25 mM HEPES-KOH pH 7.5, 100 mM KCl and 25 mM MgOAc) supplemented with 3C protease for 1 h at 4 °C.

### Purification of Rsm22-associated mtSSU intermediates

Cryo-ground yeast powder was resuspended in buffer A (25 mM HEPES-KOH pH 7.5, 100 mM KCl, 25 mM MgOAc, 0.1% Triton X-100, 2 mM DTT, PMSF, pepstatin and E64) and cleared by centrifugation at 4 °C and 40,000g for 20 min. The cleared lysate was incubated with NHS–sepharose beads (Cytiva) coupled to anti-mCherry nanobodies for 3 h at 4 °C on a nutator. Beads were pelleted by centrifugation at 4 °C for 1 min at 127g. After three washes in buffer A, the immobilized complexes were incubated with 3C protease at 4 °C for 1 h. After protease cleavage, the supernatant was applied to NHS–sepharose beads (Cytiva) coupled to anti-alfa nanobodies and incubated in buffer B (25 mM HEPES-KOH pH 7.5, 100 mM KCl and 25 mM MgOAc) for 1.5 h at 4 °C. Beads were subsequently washed three times in buffer B and the complex was eluted in the same buffer, supplemented with TEV protease.

### Purification of uS17m-associated RNA–protein complex from *PET127*-KO and wild-type yeast cells

Cryo-ground yeast powder was resuspended in buffer A (25 mM HEPES-KOH pH 7.5, 100 mM KCl, 25 mM MgOAc, 0.1% Triton X-100, 2 mM DTT, PMSF, pepstatin and E64) and cleared by centrifugation at 4 °C and 40,000g for 20 min. The cleared lysate was incubated with anti-GFP nanobody beads (Chromotek) for 3 h at 4 °C on a nutator. Beads were pelleted by centrifugation at 4 °C for 1 min at 127g. After three washes in buffer A, complexes were eluted in buffer B (25 mM HEPES-KOH pH 7.5, 100 mM KCl and 25 mM MgOAc) supplemented with TEV protease for 1 h at 4 °C.

### Cryo-EM grid preparation and data collection for yeast particles

Grids were prepared using a Vitrobot Mark IV robot (FEI Company) at 100% humidity. For the Ccm1 dataset, 3.5 µl eluted complex solution was applied to a glow-discharged Quantifoil Au R3.5/1 with a layer of 2-nm ultrathin carbon (LFH7100AR35, Electron Microscopy Sciences). The grid was incubated for 1 min inside the Vitrobot chamber before manually blotting excess solution. Of fresh sample, 3.5 µl was then reapplied before another 1-min incubation and manual blotting. A final 3.5 µl sample application and 1-min incubation were completed before adding CHAPSO to CMC (8 mM). The grid was then blotted (blot force of 8 and blot time of 8 s) and plunged into liquid ethane.

For the Rsm22 dataset, 3.5 µl eluted complex solution was applied to a glow-discharged Quantifoil Au R3.5/1 grid with a layer of 2-nm

ultrathin carbon (LFH7100AR35, Electron Microscopy Sciences). The grid was incubated for 2.5 min inside the Vitrobot chamber before adding CHAPSO to CMC (8 mM). The grid was then blotted (blot force of 8 and blot time of 3.5 s) and plunged into liquid ethane.

Grids were imaged on a Titan Krios electron microscope (FEI) with an energy filter (slit width of 20 eV) and a K3 Summit detector (Gatan) operating at 300 kV with a nominal magnification of ×64,000. SerialEM[53] was used to collect two datasets totalling 31,995 (Ccm1) and 14,111 (Rsm22) micrographs with a defocus range of −1 to −2.5 µm and a super-resolution pixel size of 0.54 Å. Micrographs contained 40 frames using a total dose of 36 e⁻ per pixel per second (specimen pixel size of 1.08 Å per pixel) with an exposure time of 2 s and a total dose of 61.73 e⁻ Å⁻². A different multi-shot strategy was used for each dataset. For the Ccm1 dataset, a multi-shot strategy was used with nine micrographs collected per hole, and nine holes were imaged at each single stage position. For the Rsm22 dataset, ten micrographs were collected per hole, and four holes were imaged at each stage position.

## Cryo-EM data processing of yeast particles

All cryo-EM processing steps were completed using RELION 3.1.1 (ref. [54]) unless noted, and are summarized in Supplementary Figs. 21–25. For each dataset, images were gain corrected, dose weighted, aligned and binned to a pixel size of 1.08 Å using the MotionCor2 implementation in RELION[55], and micrograph defocus was estimated using GCTF1.18 (ref. [56]).

For the Ccm1-associated mtSSU intermediates, an initial model for 3D classification was generated by using a test subset of 8,189 micrographs as follows: CrYOLO[57] was trained and used to pick 830,061 particles from these micrographs, and these particles were extracted at 4.32 Å per pixel (4× binning). After two rounds of 2D classification ($K = 100$ and $T = 2$), four classes were conservatively picked, which resembled different views of the mature mtSSU, resulting in 23,206 particles. These particles went through two rounds of ab initio reconstruction in cryoSPARC[60]. Refinement of 7,263 selected particles in cryoSPARC allowed generation of an ab initio initial model.

CrYOLO was trained and used to pick particles from all micrographs, resulting in 3,286,640 particles. These particles were extracted at 4.32 Å per pixel (4× binning) and split into 5 subsets with 357,328 particles each. To clean the particle stack, each subset was subjected to 3D classification with alignment ($K = 5$ and $T = 4$; 25 iterations with 7.5° angular sampling, 10 iterations with 3.7° angular sampling and 10 iterations with 1.8° angular sampling). Good classes from each 3D classification were combined and duplicates were removed, resulting in 316,817 total particles. These particles were re-extracted at 1.08 Å per pixel (1× binning), refined and subjected to Bayesian polishing and CTF refinement. A subsequent refinement resulted in a 3.13 Å consensus map. To identify conformationally distinct states in the data, two rounds of 3D classification without alignment ($K = 3$ and $T = 50$) were performed with a mask around the body domain. Two distinct classes were identified, which represented compositionally unique assembly intermediates. Both states contain a 5′ precursor bound to the mtSSU assembly factor Ccm1. 19,730 particles represent the earliest state in which the platform is disordered and a group of ribosomal proteins—which we call the backpack (bS1m, uS2m, mS23, mS37, mS42 and mS43)—that bind to the 3′ end of the 15S rRNA are absent. We refer to this intermediate as state 1. 54,519 particles represent a later state in which the platform is well ordered and backpack proteins are bound. We denote this intermediate as state 2. Each of these states were refined and subsequently processed independently.

To fully structurally characterize these intermediates, we adopted a strategy using focused classification and/or focused refinement to improve local resolutions of flexible regions, such as those occupied by assembly factors, the head and the platform. All focused refinements were performed using solvent flattening, and reported resolutions are based on the gold-standard FSC-0.143 criterion as calculated during post-processing in RELION. The local resolution of focused maps was calculated in RELION.

For the Rsm22-associated mtSSU intermediates, an initial model for 3D classification was generated by using a test subset of 3,982 micrographs as follows: CrYOLO was trained and used to pick 891,434 particles from these micrographs, and these particles were extracted at 4.32 Å per pixel (4× binning). After 2D classification ($K = 100$ and $T = 2$), 17 classes were picked, which resembled different views of the mature mtSSU, resulting in 635,817 particles. These particles went through ab initio reconstruction in cryoSPARC allowing generation of an ab initio initial model.

CrYOLO was trained and used to pick particles from all micrographs, resulting in 3,544,843 particles, which were extracted at 8.64 Å per pixel (8× binning). To clean the particle stack, all particles were subjected to 2D classification ($K = 100$ and $T = 2$) followed by 3D classification with alignment ($K = 5$ and $T = 4$; 20 iterations with 7.5° angular sampling, 10 iterations with 3.7° angular sampling and 10 iterations with 1.8° angular sampling). Good classes from this classification were combined resulting in 1,656,825 total particles. These particles were re-extracted at 1.08 Å per pixel (1× binning), refined and subjected to Bayesian polishing and CTF refinement. A subsequent refinement resulted in a 2.84 Å consensus map. It was clear from the consensus map that there was considerable flexibility in the head domain of this particle subset. To visualize this region, 3D classification without alignment ($K = 3$ and $T = 50$) was performed with a mask around the head domain. Two good classes from this state were combined and subjected to Bayesian polishing, CTF refinement and 3D refinement, resulting in a 2.65 Å map. To fully structurally characterize this intermediate, we used multibody refinement with masks around the head and body domains, resulting in two separate maps. This improved the local resolution, most notably in the head domain. Reported resolutions of these maps are based on the gold-standard FSC-0.143 criterion as calculated during post-processing in RELION. The local resolution of focused maps were calculated in RELION.

We exploited overlap in the body and head maps to generate composite maps and corresponding models that represent the predominant relative conformation between these domains for each individual state. Focused maps were fit and resampled into the overall maps in Chimera[58] and subsequently combined to create composite maps using phenix.combine_focused_maps[59]. Half-maps from each focused refinement or multibody refinement were also combined using the same procedure. The composites were then rescaled to a calibrated pixel size of 1.057 Å per pixel, and overall resolutions were calculated in RELION. cryoSPARC was used to calculate local resolution of composite maps and to generate maps filtered by local resolution. The Remote 3D-FSC Processing Server was used to calculate 3D-FSC curves for composite maps[61].

## Model building and refinement for yeast structures

For yeast states, a starting model including all ribosomal proteins and 15S rRNA (PDB: 5MRC) was used as an initial template for model building by rigid body docking into composite maps. To build assembly factors, homology models of Rsm22 and Ccm1 generated using Alphafold[64] were used as starting models before manual adjustment using COOT[62]. Changes in RNA conformation were accounted for in each state by manual refinement in COOT[62]. Finally, entire models for each state were adjusted with three cycles of refinement in PHENIX using phenix.real_space_refine using secondary structure restraints[59]. Owing to lower local resolution of Rsm22 in state 1, we trimmed side chains of this protein to $C_\beta$ after all-atom refinement, excluding residues essential for iron–sulfur cluster coordination. Model refinement statistics for yeast models can be found in Extended Data Table 2.

Maps and models were visualized in Chimera[58], ChimeraX[63] and PyMol (Schrödinger, LLC). Figures were generated using ChimeraX[63]. PyMol sessions are available for each state individually and with all states together.

## RNA extraction and northern blotting of yeast mitoribosomal RNA

Total cellular RNA was extracted from 0.2 g of frozen yeast cells after lysis by bead beating in 1 ml TRIzol (Ambion). For the analysis of pre-rRNA processing states by northern blotting, 3 µg total cellular RNA was loaded in each lane of a denaturing 1.2% formaldehyde–agarose gel (SeaKem LE, Lonza) and separated at 75 V for 3 h. After running, the separated RNA was transferred onto a cationized nylon membrane (Zeta-Probe GT, Bio-Rad) using downward capillary transfer and crosslinked to the membrane for northern blot analysis by UV irradiation at 254 nm with a total exposure of 120 mJ cm$^{-2}$ in a UV Stratalinker 2400 (Stratagene).

Before the addition of γ-32P-end-labelled DNA oligo nucleotide probes, crosslinked membranes were incubated with hybridization buffer (750 mM NaCl, 75 mM trisodium citrate, 1% (w/v) SDS, 10% (w/v) dextran sulfate and 25% (v/v) formamide) for 30 min at 65 °C. Labelled hybridization probes were incubated with the membrane first at 65 °C for 1 h and then at 37–45 °C overnight. Blotted membranes were washed once with wash buffer 1 (300 mM NaCl, 30 mM trisodium citrate and 1% (w/v) SDS) and once with wash buffer 2 (30 mM NaCl, 3 mM trisodium citrate and 1% (w/v) SDS) for 30 min each at 45 °C, before the radioactive signal was read out by exposure of the washed membranes to a storage phosphor screen, which was subsequently scanned with a Typhoon 9400 variable-mode imager (GE Healthcare). Oligonucleotide probe sequences were as follows: 15S, GTTTACTACTAGAACTACACGGGTATCG; 5′ precursor, ATTTTTTACTTTTCCTTATTATA.

## MS analysis

Purified ribonucleoprotein samples were dried and dissolved in 8 M urea, 0.1 M ammonium bicarbonate and 10 mM DTT. After reduction, cysteines were alkylated in 30 mM iodoacetamide (Sigma). Proteins were digested with LysC (endoproteinase LysC, Wako Chemicals) in less than 4 M urea followed by trypsination (trypsin gold, Promega) in less than 2 M urea. Digestions were halted by adding TFA, and digests were desalted[65] and analysed by reversed phase nano-LC–MS/MS using a Fusion Lumos (Thermo Scientific). Data were quantified and searched against the *S. cerevisiae* or *H. sapiens* Uniprot protein database (2019) concatenated with the MS2 protein sequence and common contaminations. For the search and quantitation, MaxQuant v2.0.3.0 (ref. [66]) was used. Oxidation of methionine and protein N-terminal acetylation were allowed as variable modifications and all cysteines were treated as being carbamidomethylated. The 'match between runs' option was enabled, and false discovery rates for proteins and peptides were set to 1% and 2%, respectively.

For comparative MS, protein abundances were expressed as label-free quantitation values. Data were analysed using Perseus (v1.6.10.50)[67]. In short, label-free quantitation values were log$_2$ transformed followed by a filtering requiring that a protein must be matched in all three replicates for at least one condition. A two-sided Student's *t*-test was carried out to confirm statistical significance of data.

All MS data are available in Supplementary Data 1–5.

## Western blotting

To determine whether mitochondrial translation is compromised in METTL17–GFP-tagged cells, wild-type and METTL17–GFP HEK293F cell pellets were resuspended in lysis buffer (25 mM HEPES-KOH pH 7.5, 100 mM KCl, 25 mM Mg(OAc)$_2$ and 1% Triton X-100, with 2X protease inhibitors (E64, pepstatin and PMSF)) and lysed for 30 min at 4 °C. Lysate was cleared at 17,000 *g* for 20 min at 4 °C, and total protein concentration was determined by the Bio-Rad Protein Assay Kit. Equal amounts of total protein (20 µg) was separated by SDS–PAGE and transferred to a polyvinylidene difluoride membrane before blocking in TBS-T with 5% skimmed milk for 1 h at room temperature. The membrane was then incubated with primary antibody in TBS-T with 3% skimmed milk for 2 h at room temperature at the following dilutions: MT-CO1 (clone 1D6E1A8, cat. #459600, Thermo Fisher Scientific) for 1:1,000, MT-ND1 (clone 5J5C8, cat. #MA5-42939, Thermo Fisher Scientific) for 1:1,000 and β-actin (cat. #PA1-183, Thermo Fisher Scientific) for 1:2,000. The membrane was then washed three times in TBS-T before incubation with the corresponding horseradish peroxidase (HRP)-conjugated secondary antibody: HRP-conjugated goat anti-rabbit IgG (cat. #111-035-003, Jackson ImmunoResearch) or HRP-conjugated goat anti-mouse IgG (cat. #115-035-003, Jackson ImmunoResearch) at a dilution of 1:5,000 in TBS-T with 3% skimmed milk. The membrane was then washed twice in TBS-T and once in TBS before imaging on a Typhoon 9400 imager (GE) using ECL Prime detection reagent (Amersham).

## Reporting summary

Further information on research design is available in the Nature Portfolio Reporting Summary linked to this article.

## Data availability

Raw cryo-EM micrographs have been deposited in the Electron Microscopy Public Image Archive: EMPIAR-11313. The cryo-EM maps and atomic models have been deposited in the Electron Microscopy Data Bank (EMDB) and the PDB, respectively: state A (EMD-26966 and 8CSP), state B (EMD-26967 and 8CSQ), state C (EMD-26968 and 8CSR), state D (EMD-26969 and 8CSS), state E (EMD-26970 and 8CST), state C* (EMD-26971 and 8CSU), state 1 (EMD-27249 and 8D8J), state 2 (EMD-27250 and 8D8K) and state 3 (EMD-27251 and 8D8L).

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

**Acknowledgements** We thank M. Ebrahim, J. Sotiris and H. Ng at the Evelyn Gruss Lipper Cryo-Electron Microscopy Resource Center at The Rockefeller University for assistance with grid screening and data collection; S. Mazel at the Flow Cytometry Resource Center at The Rockefeller University for assistance with single-cell sorting during human cell line generation; A. Vanden Broeck for advice with cryo-EM data processing and scripts for figures; and members of the Klinge laboratory for critical reading of this manuscript. MS data were generated by the Proteomics Resource Center at The Rockefeller University (RRID: SCR_017797) using instrumentation funded by the Sohn Conferences Foundation and the Loena M. and Harry B. Helmsley Charitable Trust, and we particularly thank H. Molina, S. Heissel and C. Peralta for assistance. This study was funded, in part, by the NIH T32 training grant 5T32GM136640-02 (to N.J.H.). and by the Robertson Foundation (to S.K.).

**Author contributions** S.K. conceived the study. N.J.H., C.B. and S.K. designed the experiments and analysed the data. N.J.H. performed all experiments in human cells, including cell line generation, complex purification, cryo-EM data processing and model building. C.B. performed all experiments in yeast cells, including strain generation, complex purification, cryo-EM data processing, model building and northern blotting. All authors wrote and edited the manuscript.

**Competing interests** The Rockefeller University has filed a patent related to the human genome editing platform used in this study, on which S.K. is named as an inventor. All other authors declare no competing interests.

**Additional information**
**Correspondence and requests for materials** should be addressed to Sebastian Klinge.

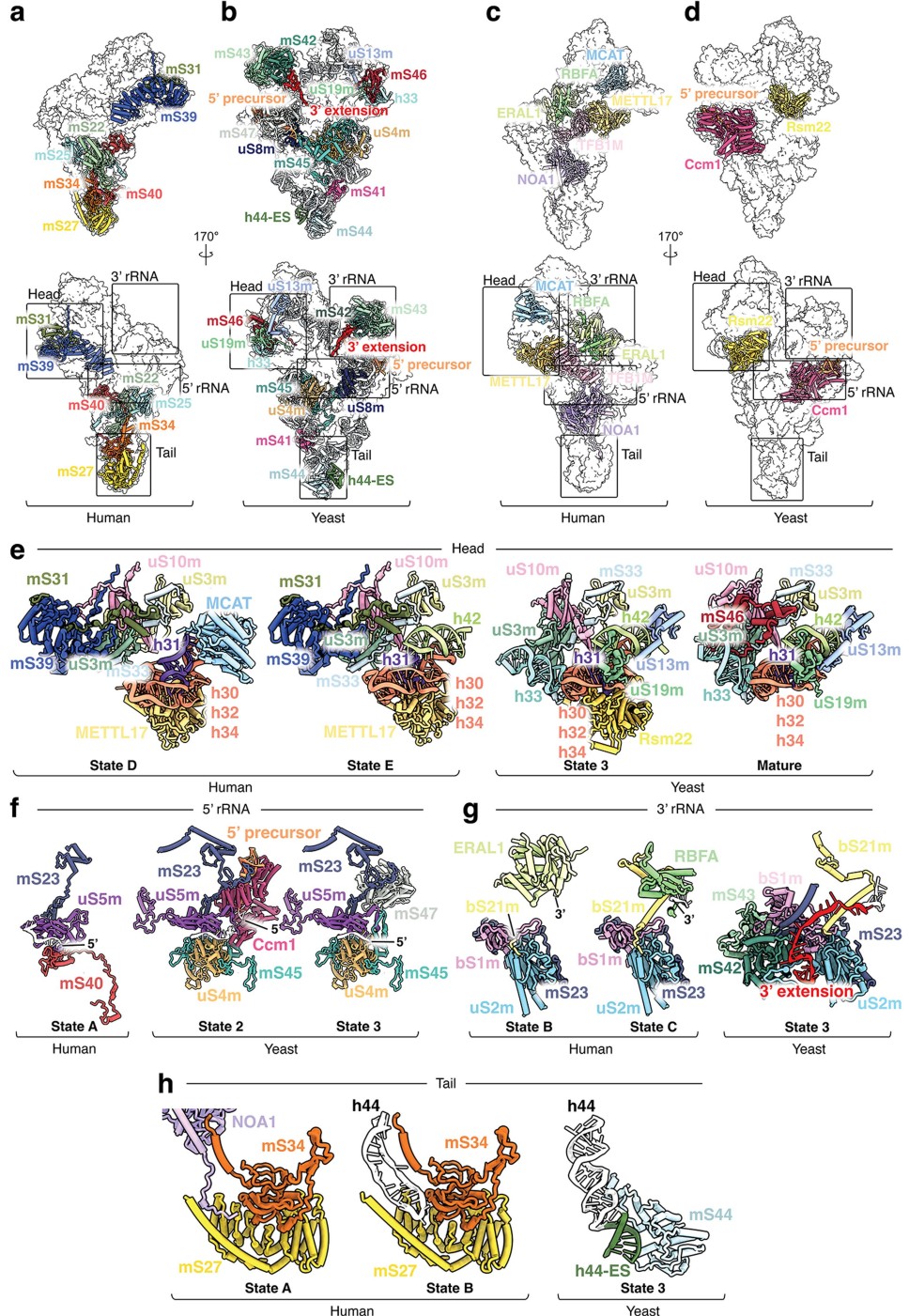

**Extended Data Fig. 1 | Roles of evolutionarily unique mitoribosomal proteins and assembly factors in mitochondrial small subunit biogenesis.** Models of: (**a**) human mitoribosomal proteins not present in yeast in context of the mature subunit (PDB:3J9M)[21] (**b**) yeast mitoribosomal proteins and rRNA not present in humans in context of the mature subunit (PDB:5MRC)[19]. Unique rRNA segments are colored white unless the region is functionally relevant within assembly states captured in this study. (**c**) human assembly factors identified in this study. (**d**) yeast assembly factors identified in this study. For panels a-d, bottom images include boxes outlining labeled regions of interest for panels e-h. (**e**) the head domain near METTL17/Rsm22 in States D and E (human) and State 3 and the mature structure (yeast, PDB:5MRC)[19]. In humans, unique protein mS31 binds unique protein mS39. In yeast, a unique rRNA segment, h33, is accommodated by unique protein mS46 after Rsm22 dissociation. Human assembly factor MCAT, which lacks a yeast homologue,

may be functionally replaced by yeast uS13m and uS19m. (**f**) proteins accommodating the 5′ rRNA in State A (human) and states 2 and 3 (yeast). The core conserved binding module formed by uS5m is augmented in yeast to integrate the 5′ precursor and Ccm1 via yeast specific proteins uS4m, mS45, and mS47. (**g**) proteins involved in 3′ stabilization in States B and C (human) and State 3 (yeast). In humans, dedicated 3′ binding assembly factors ERAL1 and RBFA stabilize the 3′ rRNA during assembly. In yeast, the 3′ rRNA is extended and is accommodated via a backpack module including unique proteins mS42, mS43, and an mS23 extension. (**h**) proteins involved in h44 docking in States A and B (human) and State 3 (yeast). Human specific proteins mS27 and mS34 allow NOA1-mediated inhibition of h44 docking, while h44 in yeast is bound by yeast protein mS44. For panels e-h, conserved proteins and rRNA segments between the two assembly systems are colored identically.

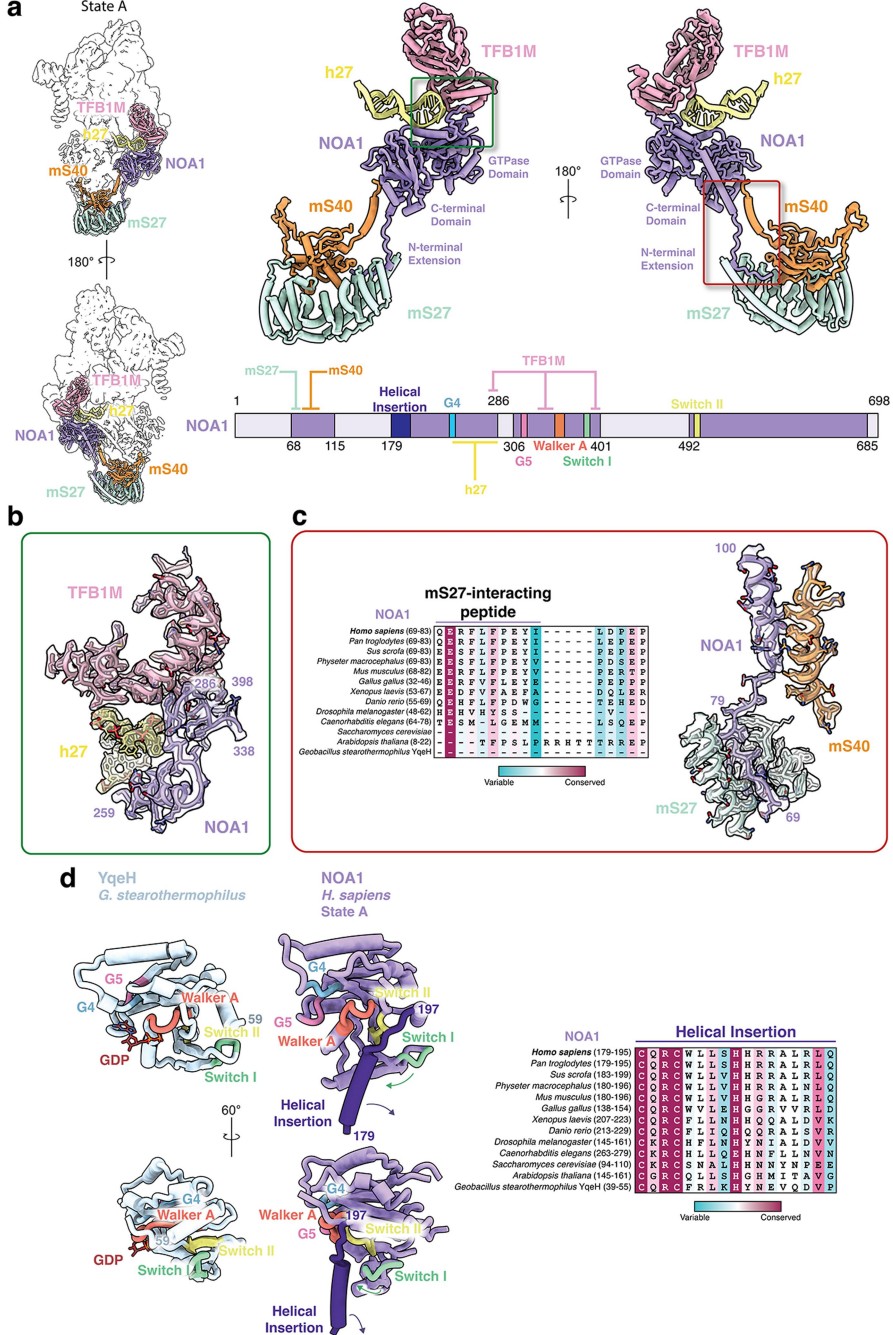

**Extended Data Fig. 2 | Details of NOA1 function within State A.** (**a**) Interaction network of NOA1 in State A in context with the entire complex (left), shown in cartoon (right), and in a schematic (bottom). (**b**) Model and cryo-EM map of region highlighted in green in (a). (**c**) Conservation analysis of NOA1 N-terminal peptide region (left) and molecular context, shown as model and cryo-EM density. (**d**) Models of the GTPase domain of related circularly permutated GTPase YqeH (left, PDB:3EC1)[38] compared to NOA1 (right). Conserved GTPase motifs are colored and labeled. Proposed rearrangements of NOA1 occurring during GTP binding are shown as arrows. Conservation analysis of the helical insertion in the GTPase domain (right).

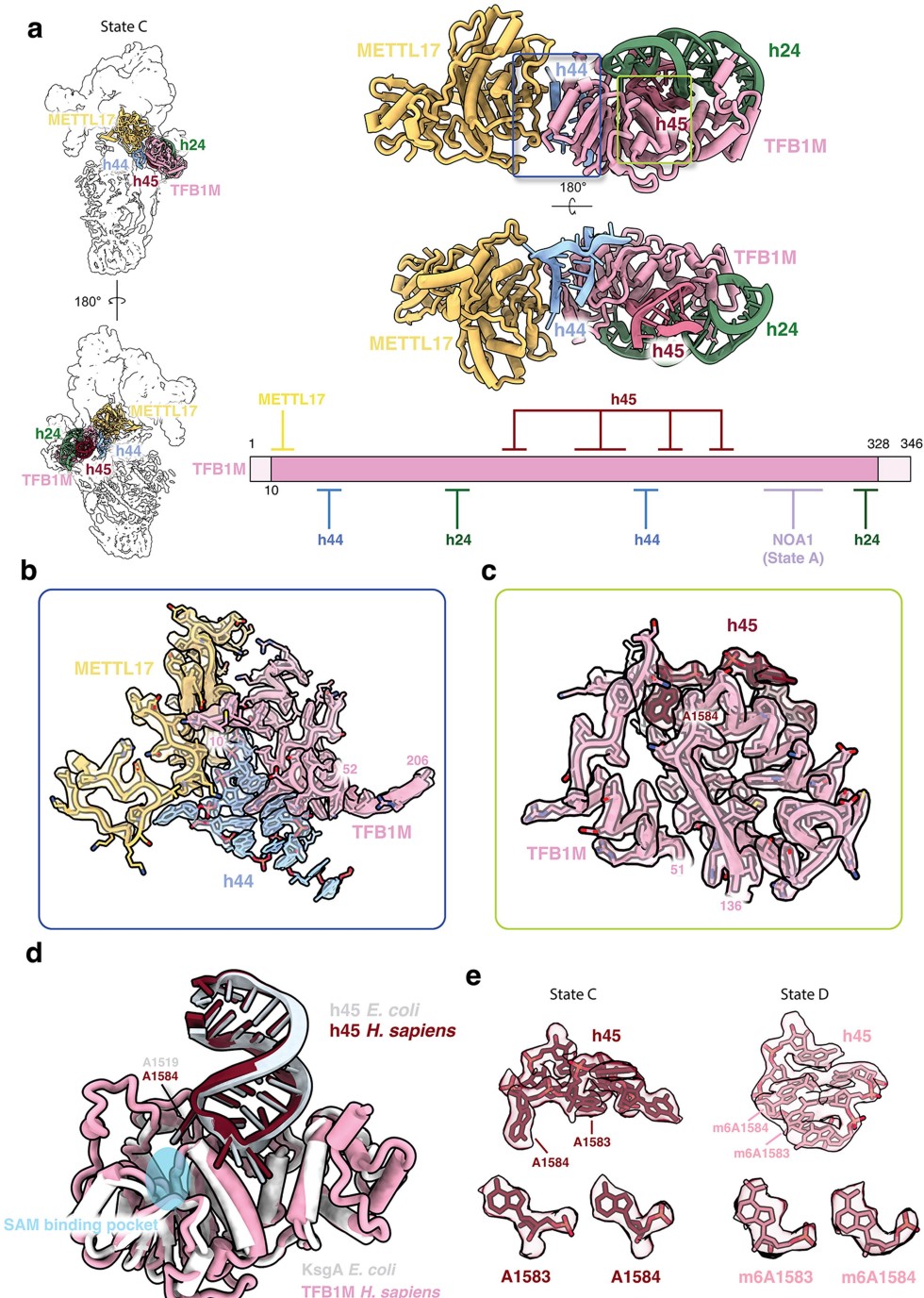

**Extended Data Fig. 3 | TFB1M plays roles in h44 maturation and h45 methylation. (a)** Interaction network of TFB1M in State C in context with the entire complex (left), shown in cartoon (right) and in a schematic (bottom). **(b)** Models and cryo-EM maps showing TFB1M h44 clamping activity. **(c)** Models and cryo-EM maps showing the enzymatic active site lacking density for methyl donor S-adenosyl methionine. **(d)** Comparison of h45 binding mode between TFB1M and its bacterial homologue KsgA (PDB:7O5H)[39]. **(e)** Comparison of h45 cryo-EM density between State C and State D, showing conformational changes and appearance of density corresponding to modified adenosine residues. Maps from both states are displayed at the same contour level.

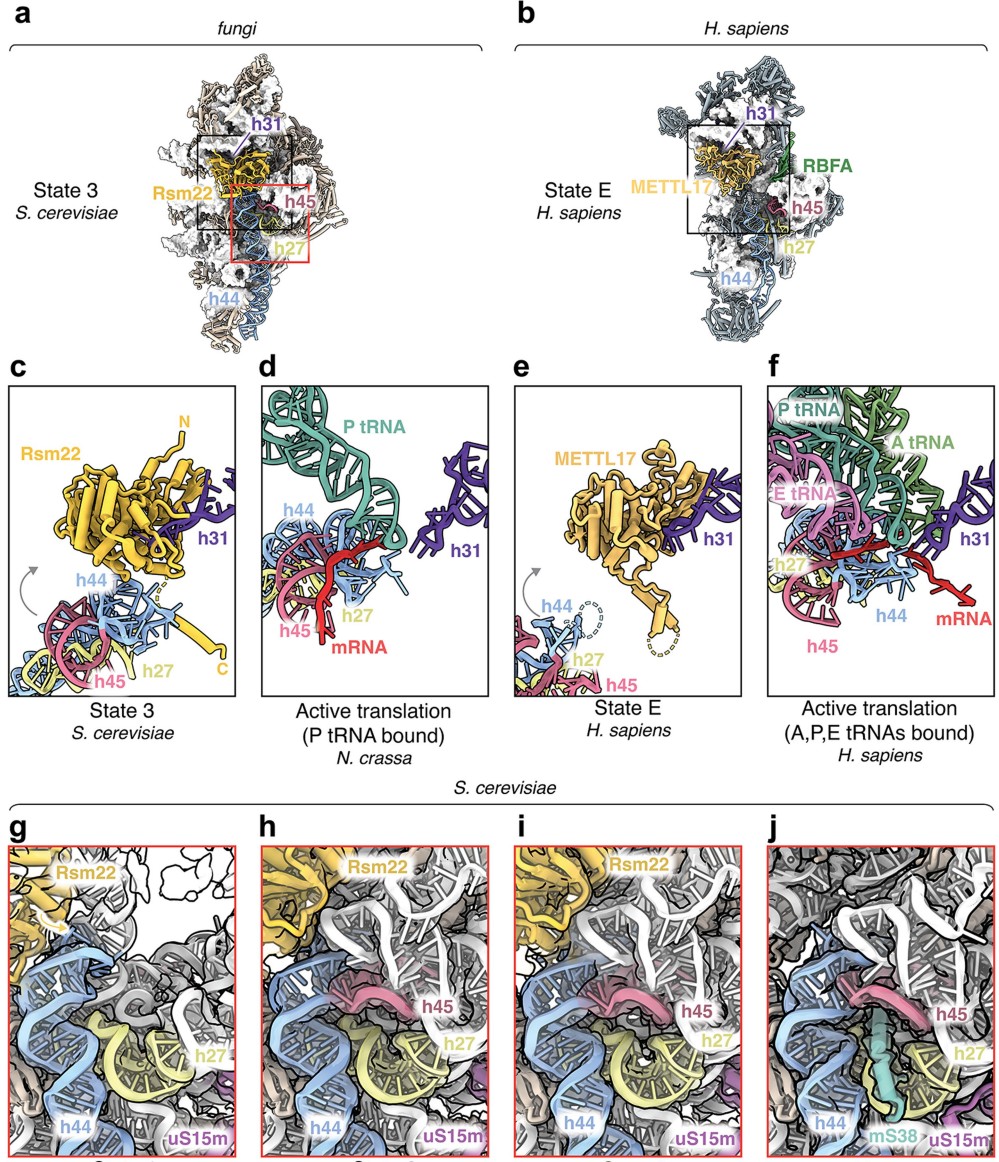

**Extended Data Fig. 4 | Conserved and distinct features of decoding center maturation in *S. cerevisiae*.** (**a**) Overall structure of yeast State 3. Black and red squares denote location of panels c and d, and g-j, respectively. (**b**) Overall structure of human State E. The black square denotes location of panels e and f. (**c**) Model of h31 bound by Rsm22, as observed in State 3. The grey arrow denotes relative movement of rRNA towards the decoding center due to subunit compaction after Rsm22 dissociation. (**d**) Same view as (c) in the structure of the *N. crassa* mitoribosome during translation (PDB:6YWY)[68], showing interactions of h31 with mRNA and the P-site tRNA. (**e**) Model of h31

bound by METTL17, as observed in State E. The grey arrow denotes relative movement of rRNA towards the decoding center due to subunit compaction after METTL17 dissociation. (**f**) Same view as (e) in the structure of the *H. sapiens* mitoribosome during translation bound to A, P, and E-site tRNAs (PDB:6ZSG)[40], showing interactions of h31 with mRNA and the P-site tRNA. In panels c-f, models are aligned on h31. (**g**–**j**) Timeline of yeast States 1–3 and the mature structure (PDB:5MRC)[19] with associated map gaussian filtered at 1.4 standard deviations overlaid. Note absence of h45 in State 1 and mS38 in assembly intermediates States 1–3.

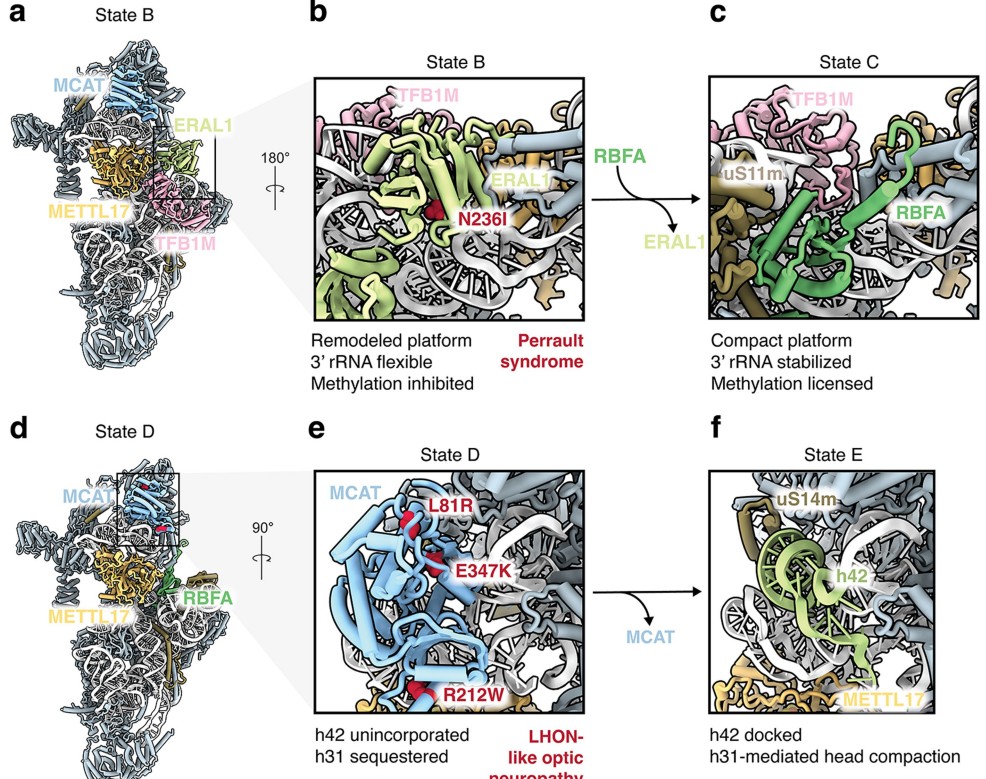

**Extended Data Fig. 5 | Structural and mechanistic context of assembly factor mutations implicated in disease.** (**a**) Model of State B highlighting the context of ERAL1 during assembly. (**b**) Location of the N236I mutant implicated in Perrault syndrome in the GTPase binding site of ERAL1[29]. (**c**) Same view as (b) in State C, showing structural transitions associated with ERAL1/RBFA exchange. (**d**) Model of State D highlighting the context of MCAT during assembly. (**e**) Location of MCAT L81R, R212W, and E347K mutations implicated in LHON-like optic neuropathy[32]. (**f**) Same view as (e) in State E, showing structural transitions associated with MCAT dissociation. In panels b, c, e, and f, properties of local rRNA elements within each state are noted.

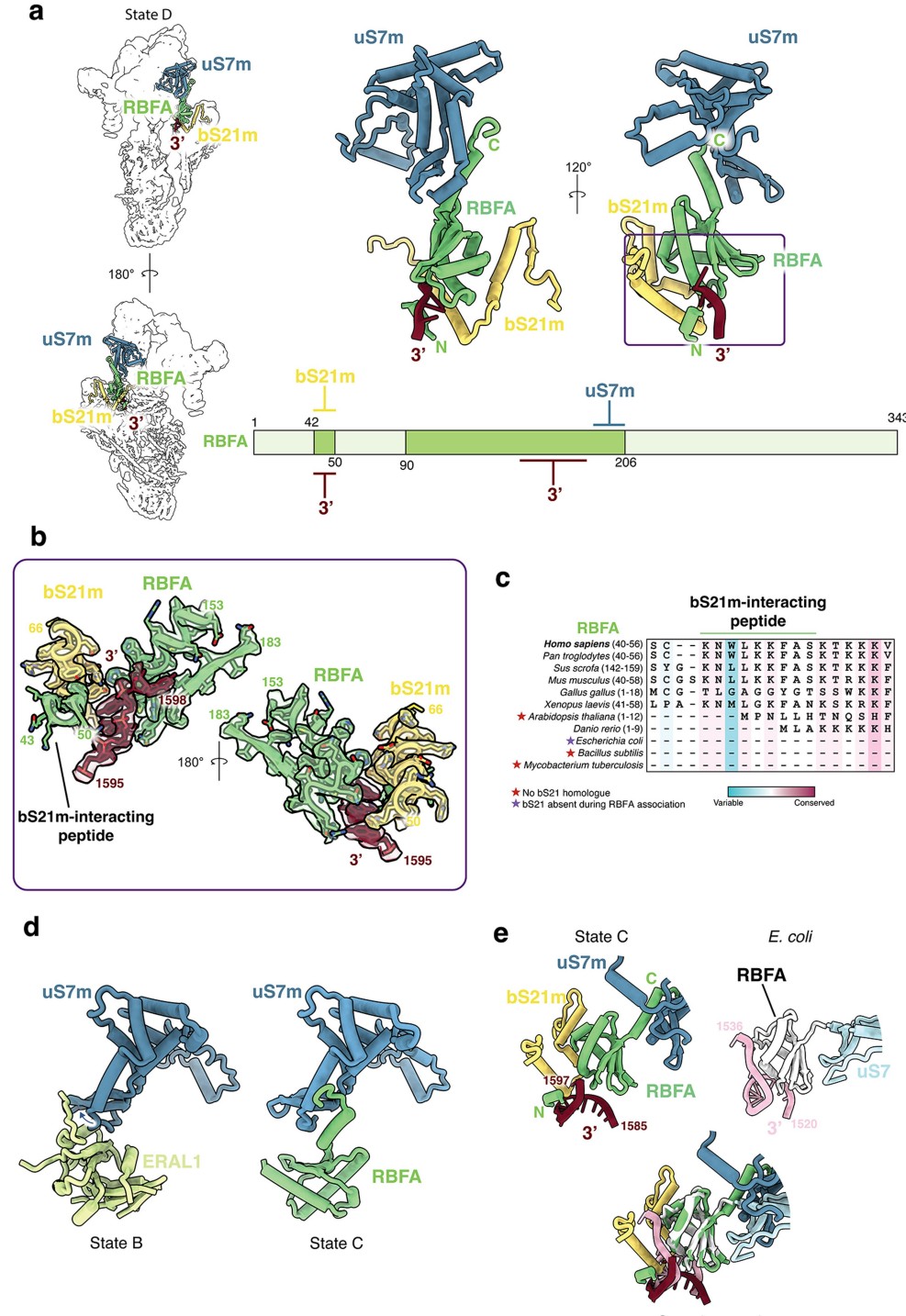

**Extended Data Fig. 6 | The 3′ rRNA binding mode of RBFA is distinct from that of *E. coli*.** (**a**) Interaction network of RBFA in State C in context with the entire complex (left), shown in cartoon (right) and in a schematic (bottom). (**b**) Model and cryo-EM map of the highlighted region in (a), showing the composite binding site formed by RBFA and bS21m. (**c**) Conservation analysis of the N-terminal peptide segment observed to interact with bS21m during 3′ rRNA binding. (**d**) Model illustrating conformational changes in uS7m during RBFA/ERAL1 exchange in States B and C. (**e**) Comparison of the 3′ rRNA binding mode between human and bacterial systems (PDB:7BOH)[23].

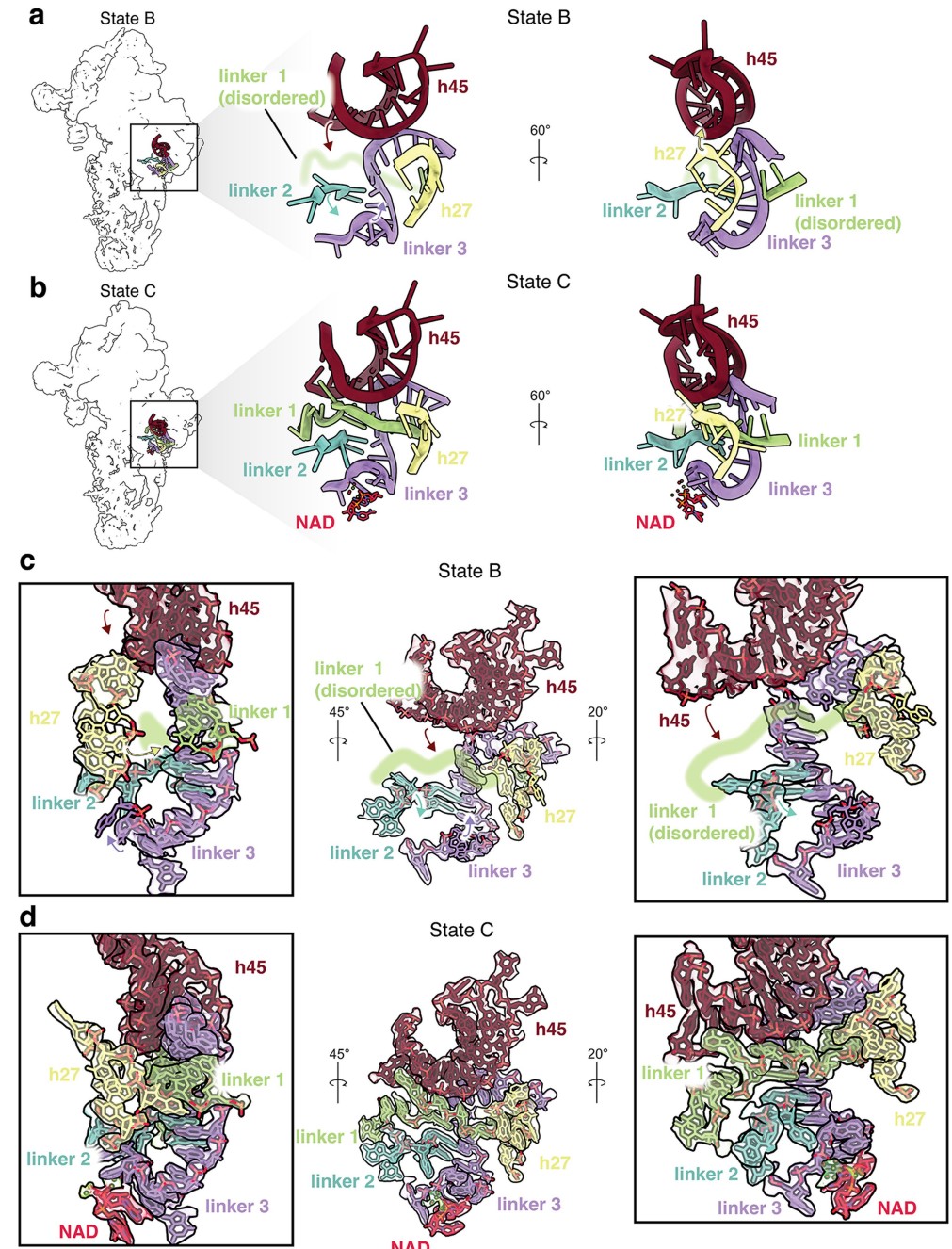

**Extended Data Fig. 7 | Molecular details of NAD binding and platform stabilization.** (**a**,**b**) rRNA elements involved in platform stabilization during the transition between state B (panel a) and state C (panel b), shown in context with the rest of the complex (left), and as cartoons (right). Movement of rRNA associated with platform stabilization shown with arrows. (**c**,**d**) Models and cryo-EM density showing transition of platform rRNA between state B (panel c) and state C (panel d). Arrows in all panels highlight conformational changes occurring during the transition from state B to state C.

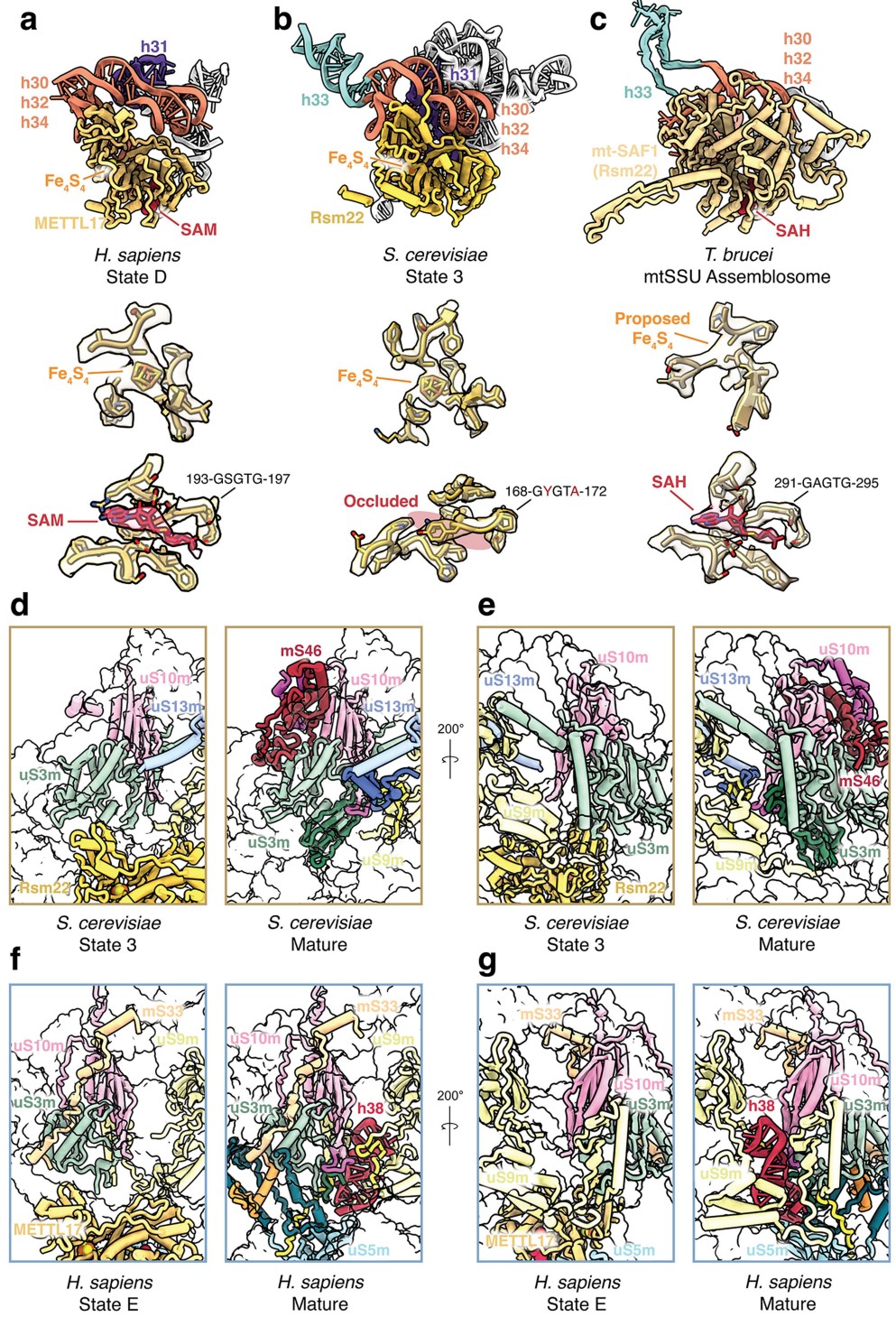

**Extended Data Fig. 8 | A conserved role of METTL17/Rsm22 in head maturation.** (**a**–**c**) Models of METTL17, Rsm22, and SAF1 (Rsm22 in *T. brucei*, PDB:6SGB)[15] respectively, with associated rRNA elements (top). Below, cryo-EM densities and models for ligands and their binding sites associated with each assembly factor (for *T. brucei*, map is EMD-10180, and empty density was originally modeled as a Zn²⁺)[15]. (**d**,**e**) Ordering of yeast mitoribosomal protein elements (shown as cartoon) in the head domain induced upon Rsm22 dissociation and head compaction. Peptides which become ordered between State 3 and the mature structure are colored in a darker shade. Additional protein and RNA components are shown as transparent surfaces. Mature structure derived from PDB:5MRC[19]. Models of RNA omitted for clarity. (**f**,**g**) Ordering of human mitoribosomal protein elements (shown as cartoon) in the head domain induced upon METTL17 dissociation and head compaction. Peptides which become ordered between State E and the mature structure are colored in a darker shade. Additional protein and RNA components are shown as transparent surfaces. Mature structure derived from PDB:6RW4[69]. Previously ordered RNA omitted for clarity in mature structures.

## Extended Data Table 1 | Cryo-EM data collection and refinement statistics of human structures

**Data Collection and Processing**

| | |
|---|---|
| Microscope | FEI Titan Krios |
| Voltage (keV) | 300 |
| Camera | Gatan K3 |
| Magnification | 64,000 |
| Pixel size at detector (Å/pixel) | 1.08 |
| Total electron exposure (e⁻/Å²) | 51 |
| Exposure rate (e⁻/pixel/sec) | 28.5 |
| Number of frames (no.) | 42 |
| Defocus range (μm) | -0.7 to -1.8 |
| Automation software | SerialEM |
| Micrographs collected (no.) | 47,037 |
| Total extracted particles (no.) | 9,109,335 |

| | State A | State B | State C | State D | State E | State C* |
|---|---|---|---|---|---|---|
| EMBD | EMD-26966 | EMD-26967 | EMD-26968 | EMD-26969 | EMD-26970 | EMD-26971 |
| PDB | 8CSP | 8CSQ | 8CSR | 8CSS | 8CST | 8CSU |
| Final particle images (no.) | 263,184 | 203,950 | 262,363 | 368,056 | 39,864 | 33,642 |
| Point-group symmetry | C1 | C1 | C1 | C1 | C1 | C1 |
| Resolution (global at FSC 0.143, Å) | 2.66 | 2.54 | 2.54 | 2.36 | 2.85 | 3.03 |
| Resolution range (local at FSC 0.143, Å) | 2.40 to 5.73 | 2.25 to 5.90 | 2.29 to 6.69 | 2.14 to 6.06 | 2.55 to 7.65 | 2.76 to 8.88 |
| Map sharpening B-factor (Å²) | -57.08 to -150.49 | -41.24 to -104.41 | -46.00 to -85.26 | -38.83 to -87.21 | -35.30 to -99.63 | -35.68 to -97.66 |
| **Model Refinement** | | | | | | |
| Refinement package (real space) | | | phenix.real_space_refine (1.19) | | | |
| Initial models used (PDB code) | | | 6RW4/2C2N/6AAX | | | |
| Model resolution cutoff (Å) | 3.0 | 2.8 | 2.8 | 2.7 | 3.2 | 3.3 |
| **Model Composition** | | | | | | |
| Non-hydrogen atoms | 69,302 | 67,706 | 69,872 | 67,466 | 66,351 | 68,663 |
| Protein residues | 6,645 | 6,265 | 6,354 | 6,062 | 5,810 | 6,119 |
| RNA residues | 738 | 803 | 854 | 847 | 889 | 882 |
| Waters | - | - | 46 | 104 | - | - |
| Ligands | Mg²⁺ 28x, FES 2x, ATP 1x, GDP 1x, K⁺ 4x, SF4 1x, Zn²⁺ 1x, SAM 1x | Mg²⁺ 27x, FES 2x, ATP 1x, GDP 1x, K⁺ 4x, SF4 1x, Zn²⁺ 1x, SAM 1x | Mg²⁺ 31x, FES 2x, ATP 1x, GDP 1x, K⁺ 5x, SF4 1x, Zn²⁺ 1x, NAD⁺ 1x, SAM 1x | Mg²⁺ 32x, FES 2x, ATP 1x, GDP 1x, K⁺ 7x, SF4 1x, Zn²⁺ 1x, NAD⁺ 1x, SAM 1x | Mg²⁺ 30x, FES 2x, ATP 1x, GDP 1x, K⁺ 4x, SF4 1x, Zn²⁺ 1x, NAD⁺ 1x, SAM 1x | Mg²⁺ 26x, FES 2x, ATP 1x, GDP 1x, K⁺ 5x, SF4 1x, Zn²⁺ 1x, NAD⁺ 1x, SAM 1x |
| **B-factors** | | | | | | |
| Protein (mean) | 37.90 | 33.14 | 38.03 | 38.72 | 38.41 | 37.93 |
| RNA (mean) | 67.53 | 52.12 | 49.58 | 50.84 | 65.93 | 67.10 |
| Ligand (mean) | 36.36 | 31.18 | 34.53 | 32.32 | 37.11 | 40.88 |
| Water (mean) | - | - | 32.30 | 30.48 | - | - |
| **RMSDs** | | | | | | |
| Bond lengths (Å) | 0.004 | 0.004 | 0.004 | 0.004 | 0.004 | 0.004 |
| Bond angles (°) | 0.643 | 0.638 | 0.914 | 0.880 | 0.902 | 0.892 |
| **Validation** | | | | | | |
| MolProbity Score | 1.07 | 1.05 | 1.05 | 1.02 | 1.04 | 1.04 |
| Clash Score | 2.78 | 2.62 | 2.65 | 2.38 | 2.58 | 2.55 |
| CaBLAM outliers (%) | 0.99 | 0.86 | 0.97 | 0.80 | 0.76 | 0.86 |
| Poor rotamers (%) | 0.00 | 0.00 | 0.00 | 0.02 | 0.00 | 0.02 |
| Cβ outliers (%) | 0.00 | 0.00 | 0.00 | 0.00 | 0.00 | 0.02 |
| EMRinger score | 3.50 | 3.59 | 3.16 | 3.45 | 3.45 | 2.72 |
| **Ramachandran Plot** | | | | | | |
| Favored (%) | 98.29 | 98.28 | 98.27 | 98.59 | 98.62 | 98.52 |
| Allowed (%) | 1.71 | 1.72 | 1.73 | 1.39 | 1.36 | 1.48 |
| Outliers (%) | 0.00 | 0.00 | 0.00 | 0.00 | 0.02 | 0.00 |
| **RNA Validation** | | | | | | |
| Average suiteness (%) | 52.1 | 54.6 | 53.8 | 57.5 | 55.0 | 53.9 |
| Good sugar puckers (%) | 99.59 | 99.25 | 99.53 | 99.64 | 99.55 | 99.32 |

**Extended Data Table 2 | Cryo-EM data collection and refinement statistics of yeast structures**

|  | Ccm1 Dataset | | Rsm22 Dataset |
|---|---|---|---|
| **Data Collection and Processing** | | | |
| Microscope | FEI Titan Krios | | FEI Titan Krios |
| Voltage (keV) | 300 | | 300 |
| Camera | Gatan K3 | | Gatan K3 |
| Magnification | 64,000 | | 64,000 |
| Pixel size at detector (Å/pixel) | 1.08 | | 1.08 |
| Total electron exposure ($e^-/Å^2$) | 61.73 | | 61.73 |
| Exposure rate ($e^-$/pixel/sec) | 36 | | 36 |
| Number of frames (no.) | 40 | | 40 |
| Defocus range (μm) | -1 to -2.5 | | -1 to 2.5 |
| Automation software | SerialEM | | SerialEM |
| Micrographs collected (no.) | 31,995 | | 14,111 |
| Total extracted particles (no.) | 3,286,640 | | 3,544,843 |

|  | **State 1** | **State 2** | **State 3** |
|---|---|---|---|
| EMBD | EMD-27249 | EMD-27250 | EMD-27251 |
| PDB | 8D8J | 8D8K | 8D8L |
| Final particle images (no.) | 19,720 | 54,519 | 381,285 |
| Point-group symmetry | C1 | C1 | C1 |
| Resolution (global at FSC 0.143, Å) | 3.81 | 3.13 | 2.59 |
| Resolution range (local at FSC 0.143 Å) | 3.0-31.5 | 2.8-16.4 | 2.24-6.2 |
| Map sharpening B-factor ($Å^2$) | -44.41 to -168.34 | -43.59 to -54.33 | -40.99 to -45.77 |
| **Model Refinement** | | | |
| Refinement package (real space) | phenix.real_space_refine (1.19) | | |
| Initial models used (PDB code) | 5MRC | | |
| Model resolution cutoff (Å) | 4.0 | 3.7 | 2.8 |
| **Model Composition** | | | |
| Non-hydrogen atoms | 51,386 | 88,977 | 87,007 |
| Protein residues | 3,402 | 6,864 | 6,734 |
| RNA residues | 1183 | 1538 | 1506 |
| Waters | 0 | 3 | 3 |
| Ligands | $Mg^{2+}$ 29x, SF4 1x | $Mg^{2+}$ 72x, ATP 1x, SF4 1x | $Mg^{2+}$ 72x, ATP 1x, SF4 1x |
| **B-factors** | | | |
| Protein (mean) | 59.05 | 26.89 | 15.10 |
| RNA (mean) | 89.90 | 45.43 | 22.86 |
| Ligand (mean) | 31.14 | 34.36 | 19.41 |
| Water (mean) | - | 18.16 | 8.96 |
| **RMSDs** | | | |
| Bond lengths (Å) | 0.004 | 0.004 | 0.005 |
| Bond angles (°) | 0.842 | 0.853 | 0.902 |
| **Validation** | | | |
| MolProbity Score | 1.23 | 1.10 | 1.18 |
| Clash Score | 4.13 | 3.12 | 2.68 |
| CaBLAM outliers (%) | 1.01 | 0.74 | 1.04 |
| Poor rotamers (%) | 0.04 | 0.05 | 0.00 |
| Cβ outliers (%) | 0.00 | 0.00 | 0.00 |
| EMRinger score | 2.35 | 2.77 | 4.16 |
| **Ramachandran Plot** | | | |
| Favored (%) | 97.85 | 98.26 | 97.33 |
| Allowed (%) | 2.15 | 1.74 | 2.67 |
| Outliers (%) | 0.00 | 0.00 | 0.00 |
| **RNA Validation** | | | |
| Average suiteness (%) | 50.3 | 53.3 | 53.3 |
| Good sugar puckers (%) | 98.99 | 99.54 | 98.67 |

# Reporting Summary

## Statistics

For all statistical analyses, confirm that the following items are present in the figure legend, table legend, main text, or Methods section.

| n/a | Confirmed | |
|---|---|---|
| ☐ | ☒ | The exact sample size (*n*) for each experimental group/condition, given as a discrete number and unit of measurement |
| ☐ | ☒ | A statement on whether measurements were taken from distinct samples or whether the same sample was measured repeatedly |
| ☐ | ☒ | The statistical test(s) used AND whether they are one- or two-sided *Only common tests should be described solely by name; describe more complex techniques in the Methods section.* |
| ☒ | ☐ | A description of all covariates tested |
| ☐ | ☒ | A description of any assumptions or corrections, such as tests of normality and adjustment for multiple comparisons |
| ☐ | ☒ | A full description of the statistical parameters including central tendency (e.g. means) or other basic estimates (e.g. regression coefficient) AND variation (e.g. standard deviation) or associated estimates of uncertainty (e.g. confidence intervals) |
| ☐ | ☒ | For null hypothesis testing, the test statistic (e.g. *F*, *t*, *r*) with confidence intervals, effect sizes, degrees of freedom and *P* value noted *Give P values as exact values whenever suitable.* |
| ☒ | ☐ | For Bayesian analysis, information on the choice of priors and Markov chain Monte Carlo settings |
| ☒ | ☐ | For hierarchical and complex designs, identification of the appropriate level for tests and full reporting of outcomes |
| ☒ | ☐ | Estimates of effect sizes (e.g. Cohen's *d*, Pearson's *r*), indicating how they were calculated |

*Our web collection on statistics for biologists contains articles on many of the points above.*

## Software and code

Policy information about availability of computer code

| Data collection | Cryo-EM datasets were collected using SerialEM (v. 3.8) on an FEI Titan Krios (FEI/Thermofisher) microscope using an energy filter at a slit width of 20 eV operating at 300 kV and 64,000x magnification. 3 datasets were collected at super-resolution pixel size of 0.54 Å/px, with defocus values ranging from -0.7 to -2.5 μm. Electron dose ranged between 51 and 62 electrons/Å2 over 40-42 frames. For comparative IP-MS/MS experiments, samples were analyzed by reversed phase nano-LC-MS/MS using a Fusion Lumos (Thermo Scientific). |
|---|---|
| Data analysis | Cryo-EM movies were aligned and averaged in Relion (v. 3.1.0) using the MotionCor2-like algorithm implemented in Relion , and Contrast transfer function parameters were estimated using GCTF1.18. Particles were picked using crYOLO (v. 1.7.6) and Relion was used for subsequent classification and refinement steps. Models were manually built in Coot (v. 0.9) using starting models from the PDB (6RW4, 2C2N,6AAX,5MRC) and from the AlphaFold database. Refinement was performed using PHENIX 1.19. Final models were validated using MolProbity, EMRinger, and phenix.rna_validate in PHENIX 1.19. For comparative IP-MS/MS experiments, Data were quantified and searched against the S. cerevisiae or H. sapiens Uniprot protein database (2019) concatenated with the MS2-protein sequence and common contaminations. For the search and quantitation, MaxQuant v. 2.0.3.0 was used. Oxidation of methionine and protein N-terminal acetylation were allowed as variable modifications and all cysteines were treated as being carbamidomethylated. The 'match between runs' option was enabled, and false discovery rates for proteins and peptides were set to 1% and 2% respectively. Protein abundances were expressed as LFQ (label free quantification) values. Data were analyzed using Perseus (v.1.6.10.50). |

For manuscripts utilizing custom algorithms or software that are central to the research but not yet described in published literature, software must be made available to editors and reviewers. We strongly encourage code deposition in a community repository (e.g. GitHub). See the Nature Portfolio guidelines for submitting code & software for further information.

## Data

Atomic coordinates and EM maps have been deposited in the Protein Data Bank and Electron Microscopy Data Bank under the following accession codes: State A (EMDB-26966, PDB 8CSP), State B (EMD-26967, PDB 8CSQ), State C (EMD-26968, PDB 8CSR), State D (EMD-26969, PDB 8CSS), State E (EMD-26970, PDB 8CST), State C* (EMD-26971, PDB 8CSU), State 1 (EMD-27249, PDB 8D8J), State 2 (EMD-27250, PDB 8D8K), State 3 (EMD-27251, PDB 8D8L). Raw cryo-EM micrographs for each state have been deposited to EMPIAR (EMPIAR-11313).
Data used but not generated in this study: Starting models for model building: human mtSSU-IF3 (PDB 6RW4), human MCAT (PDB 2C2N), human TFB1M (PDB 6AAX), yeast mitoribosome (PDB 5MRC). Search database for IP-MS/MS experiments: Uniprot protein database (2019). Antibodies used during cell line generation: nanobody against E. coli BtuF (PDB 5OVW), nanobody against HIVp24 (PDB 5O2U).

## Human research participants

Policy information about studies involving human research participants and Sex and Gender in Research.

| | |
|---|---|
| Reporting on sex and gender | N/A |
| Population characteristics | N/A |
| Recruitment | N/A |
| Ethics oversight | N/A |

Note that full information on the approval of the study protocol must also be provided in the manuscript.

# Field-specific reporting

Please select the one below that is the best fit for your research. If you are not sure, read the appropriate sections before making your selection.

☒ Life sciences ☐ Behavioural & social sciences ☐ Ecological, evolutionary & environmental sciences

For a reference copy of the document with all sections, see nature.com/documents/nr-reporting-summary-flat.pdf

# Life sciences study design

All studies must disclose on these points even when the disclosure is negative.

| | |
|---|---|
| Sample size | Sample sizes for cryo-EM datasets were not predetermined. 47,037 (Human dataset), 31,995 (Yeast Ccm1 dataset), and 14,111 (Yeast Rsm22 dataset) micrographs were collected. The number of particles extracted from these micrographs were not predetermined. For comparative IP-MS/MS experiments, one yeast strain was used for three separate purification and mass spectrometry experiments (biological triplicate). |
| Data exclusions | Mis-aligned, damaged, or contaminating particles were excluded from final reconstructions during image processing. For comparative IP-MS/MS experiments, data were filtered such that a protein must be present in all 3 replicates for at least 1 condition. |
| Replication | Immunoprecipitation experiments for complex purification and structural analysis were repeated at least 3 times with similar results. Data for comparative IP-MS/MS experiments were gathered from three purification experiments, and three out of three experiments showed similar results. Western blotting experiments were performed three times with similar results for all experiments. Northern blotting experiments were performed twice with similar results. |
| Randomization | During refinement, the gold-standard approach was used to randomly assign particles to half-sets of data that are independently averaged and compared to obtain resolution estimates. |
| Blinding | During single particle analysis, particles are randomly assigned into half-sets, thus no blinding is applicable. |

# Reporting for specific materials, systems and methods

We require information from authors about some types of materials, experimental systems and methods used in many studies. Here, indicate whether each material, system or method listed is relevant to your study. If you are not sure if a list item applies to your research, read the appropriate section before selecting a response.

## Materials & experimental systems

| n/a | Involved in the study |
|---|---|
| ☐ | ☒ Antibodies |
| ☐ | ☒ Eukaryotic cell lines |
| ☒ | ☐ Palaeontology and archaeology |
| ☒ | ☐ Animals and other organisms |
| ☒ | ☐ Clinical data |
| ☒ | ☐ Dual use research of concern |

## Methods

| n/a | Involved in the study |
|---|---|
| ☒ | ☐ ChIP-seq |
| ☐ | ☒ Flow cytometry |
| ☒ | ☐ MRI-based neuroimaging |

## Antibodies

| | |
|---|---|
| Antibodies used | MT-CO1 (Thermo Fisher Scientific, Cat. #459600, Clone 1D6E1A8), MT-ND1 (Thermo Fisher Scientific, Cat. #MA5-42939, Clone 5J5C8), Beta actin (Thermo Fisher Scientific, Cat. #PA1-183), HRP-conjugated goat anti-rabbit IgG (Jackson ImmunoResearch, Cat. #111-035-003), HRP-conjugated goat anti-mouse IgG (Jackson ImmunoResearch, Cat. #115-035-003). |
| Validation | All antibodies used are commercially available and validated by the manufacturer. For MT-CO1, the antibody was verified by cell treatment to ensure that the antibody binds to the antigen stated. |

## Eukaryotic cell lines

Policy information about cell lines and Sex and Gender in Research

| | |
|---|---|
| Cell line source(s) | FreeStyle 293-F cells were obtained from ThermoFisher Scientific. |
| Authentication | FreeStyle 293-F cells were purchased from ThermoFisher Scientific are specially adapted to grow in suspension culture in FreeStyle 293 expression medium. Cells were grown accordingly and matched producer specifications. Cells were not otherwise validated. |
| Mycoplasma contamination | Cell lines were not tested for mycoplasma. |
| Commonly misidentified lines (See ICLAC register) | No commonly misidentified lines were used in the study. |

## Flow Cytometry

### Plots

Confirm that:

☒ The axis labels state the marker and fluorochrome used (e.g. CD4-FITC).

☒ The axis scales are clearly visible. Include numbers along axes only for bottom left plot of group (a 'group' is an analysis of identical markers).

☒ All plots are contour plots with outliers or pseudocolor plots.

☒ A numerical value for number of cells or percentage (with statistics) is provided.

### Methodology

| | |
|---|---|
| Sample preparation | Cells were washed in 1X PBS with 0.1% BSA before incubation with fluorescent nanobodies corresponding to cell surface epitopes expressed on transfected cells. Cells were subsequently washed and resuspended in 1X PBS with 0.1% BSA before filtering to remove cell clumps and sorting via FACS. |
| Instrument | BD FACSAria cell sorter (BD Biosciences) |
| Software | FACSDiva (BD Biosciences) |
| Cell population abundance | The final population of sorted cells was 0.6% of the starting population. These cells were subsequently analyzed using PCR to confirm genomic integration of tags. |
| Gating strategy | Cells were first sorted to identify living cells using DAPI staining and isolation of FSC/SSC singlets before sorting for GFP positive cells displaying higher fluorescence than the majority of the cell population. Subsequently, these cells were sorted based on the presence of fluorescence from both fluorophore conjugated nanobodies binding to the cell surface. |

☒ Tick this box to confirm that a figure exemplifying the gating strategy is provided in the Supplementary Information.

