## [Peer Review File · Nature]

Manuscript Title: Principles of mitoribosomal small subunit assembly in eukaryotes

Reviewer Comments & Author Rebuttals

Reviewer Reports on the Initial Version:

Referees' comments:

Referee #1 (Remarks to the Author):

This comprehensive study by Harper et al. provides high-resolution structures of 9 assembly intermediates of the human and yeast small mitoribosomal subunit. They purified human mtSSU complexes via endogenous tagged METTL17, a so far poorly investigated methyltransferase, and solved the structures of 6 assembly states with 6 associated biogenesis factors: METTL17, ERAL1, TFB1M, NOA1, RBFA, and MCAT, which has not been linked to mitoribosome assembly so far. Similarly, they isolated yeast mtSSU intermediates using Ccm1 and Rsm22 (METTL17 homolog) as baits and present 3 biogenesis states. The obtained structures enable the authors to suggest a sequential assembly scheme of the mtSSU and to explain the molecular function of involved factors. In addition, the comparison between human and yeast allows conclusions regarding evolutionary conserved mechanisms and species-specific adaptation during mtSSU maturation. The data is of high quality and the manuscript is well written. This study will be of high interest to a broad readership, but especially in the field of mitochondrial ribosome biogenesis

I have only a few comments/concerns, which the authors may want to address:

1. Did the authors check whether GFP tagged biogenesis factors like human METTL17 or yeast Rsm22 and Ccm1 are functional, meaning whether translationally active mitoribosomes are formed, e.g. by [35S]Methionine de novo mitochondrial translation or sucrose gradient centrifugation? GFP is a relatively large tag and might interfere with the function of METTL17. I think it is important to confirm the functionality of the tagged protein to exclude possible stalling events during mtSSU biogenesis which might raise certain assembly intermediates.
2. The main text or discussion could be extended to put the results more into the context of the previous study by Itoh et al. (2022); e.g. do the authors see all ribosomal proteins including mS37/CHCHD1? It is present in their LC-MS/MS list, but according to Itoh et al. it is the last ribosomal protein assembling to the mtSSU at a very late stage downstream of RBFA.
3. Do the authors have any indication whether METTL17 act as a RNA methyltransferase? It has been proposed that METTL17 is required for 12S rRNA methylation at position 1486 (m4C1486) like METTL15 and at position 1488 (m5C1488) like NSUN4. However, according to the study by Harper et al. it might be that reduced C1486 and C1488 methylation in METTL17-deficient cells is a secondary effect as the function of METTL17 might be a prerequisite for METTL15 action. Maybe the authors can include a short comment in the main text.

Minor points:

Abstract and main text (p.2-3, 16)

The mitochondrial genome also encodes for components of the ATP synthase and not only for respiratory chain complexes. Thus, it would be better to use the term “oxidative phosphorylation” instead of “respiratory chain” (line 27/50/362).

References 6-9 (line 57) refer to structural studies of the mitochondrial ribosome and not to human diseases and mutations in ribosomal proteins. The authors should use more suitable references. Delete hyphen in “mt-SSU” to be consistent (line 195).

Referee #2 (Remarks to the Author):

The manuscript by Harper et al. reports the results of a structural investigation by cryo-EM of the assembly of the mitochondrial ribosome (mitoribosome) from yeast and human. It details several intermediate stages of maturation for both the mitochondrial small ribosomal subunits (mtSSU). Among the most interesting findings, the authors describe the involvement of GTPases in the maturation at early stage of the mtSSU decoding center and rRNA processing and folding. Thanks to the parallel investigation of the mtSSU maturation from both yeast and human, the authors are able to unravel several conserved and species-specific maturation steps. Thus, the authors derived several “fundamental principles” for the mtSSU maturation: “1- stepwise activities of molecular switches control early rRNA folding events that lay a foundation for the formation of a functional decoding center”. “2- integration of the rRNA 3’ end facilitates compaction of the functional rRNA core, with yeast and human systems each having evolved distinct solutions to do so”. “3- recognition and processing of 5’ pre-rRNA is a unique feature of the yeast mtSSU assembly pathway, and is carried out by distinct assembly factors”. “4- a conserved assembly factor orchestrates maturation of the head domain and prevents premature engagement of mRNA and the mtLSU”.

In addition to the novel findings, the manuscript details several interesting methodological approaches, such as the use of CRISPR-based biallelic tagging to generate stable HEK293F cells endogenously expressing assembly factor. Such method can be applied for the studies of other factors/proteins involved in a plethora of biological functions.

Finally, the supplementary figures present a wealth of experimental procedures and workflows and will be surely of use for numerous research groups.

The paper is of quality, well written and very clear, the figures are aesthetic and mostly clear (I have few comments on the labeling of the figures), the references appear to be sufficient and the methods are appropriately detailed. One could appreciate the great complementarity this work offers to the recent work from the Amunts lab (Itoh et al., Nature 2022) detailing the maturation of the human mtSSU at a later stage, in addition to the transition from maturation to the translation initiation process.

As the abundant supplementary information addressed all of my technical questions, I only have minor comments and suggestions to improve the quality of what is already a very smooth manuscript.

The authors write in the abstract “high-resolution structures”. This term is awkward, what is “high-resolution”? better than 3 Å? Up to 2.5 Å? if better than 2 Å, is it still “high-resolution”? It is highly

preferable that the authors remove this term and simply state a range of resolution (from 3 to 2.4 Å).

Figure 3, panels A to D, some labels are difficult to read because they are colorful and superimposed to the structure. I suggest to increase the white shadow below or to simply write in black and use arrows instead.

Figure 4, Panels A to D, similar comment to Figure 3.

Referee #3 (Remarks to the Author):

The manuscript from Harper et al. is an outstanding and comprehensive study on the maturation of the small ribosomal subunit. The molecular mechanisms described in the manuscript on how the combined action of assembly factors mature the critical functionally regions of the small subunit will transform our understanding of this process. This manuscript provides, in my view, for the first time a comprehensive description of how these regions are assembled in a coordinated manner. I certainly see this manuscript having the impact, novelty and technical quality expected by readers at Nature. I support his publication in Nature after the authors consider my comments below that may add more clarity and relevance to the study.

The authors determine nine high-resolution structures of in vivo assembled small subunit assembly intermediates from yeast and human mitochondria and use them to describe the maturation process of the decoding center, platform region, processing of the 5' end of the 15S rRNA molecule and head domain. These are extremely complex processes with multiple assembly factors, rRNA helices and r-proteins often acting, changing their conformation, and simultaneously interacting. The authors did an outstanding job in the design of the figures. They significantly assist in understanding the accompanying description of the mechanisms in the text and make the paper easy to follow.

A strength of the study is the comparative done for the maturation process between human mitochondria and yeast.

I found particularly intriguing the description of the maturation mechanism for the decoding center. GTPase NOA1, methyltransferase TFB1M (homologue to bacterial KsgA) and RBFA control the maturation of the decoding center. This includes the docking of helix 44, mainly regulated by NOA1, and positioning of helices 44, 27 and 45 after NOA1 and TFB1M leave to create the binding site for mS38. While I agree with the interpretation of the structures, the chronologic description of events described in the manuscript is presented as the only possible sequence of events. Similarly, the maturation process described here for the head, platform and processing of the 5' end, all processes are described as occurring in a strictly linear fashion.

Whereas the cryo-EM structures likely describe the most frequent or 'canonical' maturation pathway, it is well established that ribosome biogenesis occurs through multiple parallel assembly pathways. That concept is completely absent from the mechanistic descriptions in this manuscript. All is described in a strict chronology of events that occur in a linear fashion. I believe this concept should be integrated across the entire manuscript and in interpreting all the maturation events described for each region.

A second point I would like the authors to clarify is the criteria used for the chronological arrangement of the structures obtained for the assembly intermediates. I see the absence and presence of assembly factors, as well as the observed protein complement in the intermediates, as the data that guide the authors in the placement of the intermediates in the linear timeline. This interpretation of the structures assumes a somehow 'simplistic' view of how assembly factors and r-proteins promote assembly. It assumes factors and ribosomal proteins bind once, act and either leave (factors) or stay (r-proteins). Is it possible that some of the factors may need to bind and be released multiple times to catalyze the folding of a particular site? Is it possible a factor may act at different times in the linear timeline and catalyze various steps on the folding of a region? Is it possible these two events catalyzed by the same factor may occur at different times because they are done through functional interplays with other assembly factors?

It is likely that if any of these types of events are occurring (parallel pathways of assembly and factors acting multiple times), the nine presented structures may not be sufficient to provide a complete description of the maturation events. This is undoubtedly a limitation of this study that should be acknowledged by the authors as part of the discussions. As of written, the manuscript describes the maturation process of the small ribosomal subunit as a process occurring in a strict sequence of events. This model does not align well with our current understanding of the ribosome assembly process as a flexible and plastic process capable to occur through multiple pathways in which maturation steps do not necessarily occur in a strict order.

Author Rebuttals to Initial Comments:

Response to the reviewers

Referees' comments:

Referee #1 (Remarks to the Author):

This comprehensive study by Harper et al. provides high-resolution structures of 9 assembly intermediates of the human and yeast small mitoribosomal subunit. They purified human mtSSU complexes via endogenous tagged METTL17, a so far poorly investigated methyltransferase, and solved the structures of 6 assembly states with 6 associated biogenesis factors: METTL17, ERAL1, TFB1M, NOA1, RBFA, and MCAT, which has not been linked to mitoribosome assembly so far. Similarly, they isolated yeast mtSSU intermediates using Ccm1 and Rsm22 (METTL17 homolog) as baits and present 3 biogenesis states. The obtained structures enable the authors to suggest a sequential assembly scheme of the mtSSU and to explain the molecular function of involved factors. In addition, the comparison between human and yeast allows conclusions regarding evolutionary conserved mechanisms and species-specific adaptation during mtSSU maturation. The data is of high quality and the manuscript is well written. This study will be of high interest to a broad readership, but especially in the field of mitochondrial ribosome biogenesis

We thank reviewer 1 for the positive assessment of our work.

I have only a few comments/concerns, which the authors may want to address:

1. Did the authors check whether GFP tagged biogenesis factors like human METTL17 or yeast Rsm22 and Ccm1 are functional, meaning whether translationally active mitoribosomes are formed, e.g. by [³⁵S]Methionine de novo mitochondrial translation or sucrose gradient centrifugation? GFP is a relatively large tag and might interfere with the function of METTL17. I think it is important to confirm the functionality of the tagged protein to exclude possible stalling events during mtSSU biogenesis which might raise certain assembly intermediates.

We agree with reviewer 1 that interference of GFP with biological function must be minimized. Therefore, in yeast, a 40 amino acid linker was used between Rsm22 and GFP so that GFP is located approximately 100 Angstroms away from the maturing mtSSU. To ensure that mitoribosome assembly and function were not compromised, all yeast strains were grown in glycerol as outlined in Materials and Methods. The linker used for human mtSSU assembly intermediates was even larger, spanning approximately 130 amino acids so that GFP is located approximately 300 Angstroms away from mtSSU assembly intermediates and steric interference can essentially be ruled out.

2. The main text or discussion could be extended to put the results more into the context of the previous study by Itoh et al. (2022); e.g. do the authors see all ribosomal proteins including mS37/CHCHD1? It is present in their LC-MS/MS list, but according to Itoh et al. it is the last ribosomal protein assembling to the mtSSU at a very late stage downstream of RBFA.

Since our purification of mtSSU assembly intermediates in human cells include an initial isolation of mitochondria, we believe that very low levels of mature mitoribosomes present in our sample are the reason why mS37 is seen in our mass spectrometry data.

3. Do the authors have any indication whether METTL17 act as a RNA methyltransferase? It has been proposed that METTL17 is required for 12S rRNA methylation at position 1486 (m4C1486) like METTL15 and at position 1488 (m5C1488) like NSUN4. However, according to the study by Harper et al. it might be that reduced C1486

and C1488 methylation in METTL17-deficient cells is a secondary effect as the function of METTL17 might be a prerequisite for METTL15 action. Maybe the authors can include a short comment in the main text.

We thank reviewer 1 for this comment and indeed believe that METTL17 is not an active RNA methyltransferase such that its depletion likely causes secondary effects. To address this in the text, we have added the following statement on pages 14/15:

“The binding of METTL17 to h31 also prevents association of the late-stage assembly factor METTL15, which is only observed in assembly intermediates with a mature head domain²¹. These events enforce a chronology of human mtSSU assembly in which head formation and compaction precedes final METTL15-mediated modification of the decoding center. The architectural role of METTL17 and position in the assembly pathway upstream of METTL15 would further explain rRNA methylation defects observed in METTL17-deficient cells⁴⁷.”

Reference 47:

Shi, Z. *et al.* Mettl17, a regulator of mitochondrial ribosomal RNA modifications, is required for the translation of mitochondrial coding genes. *FASEB J* **33**, 13040–13050 (2019).

Minor points:

Abstract and main text (p.2-3, 16) The mitochondrial genome also encodes for components of the ATP synthase and not only for respiratory chain complexes. Thus, it would be better to use the term “oxidative phosphorylation” instead of “respiratory chain” (line 27/50/362).

We agree with reviewer 1 that the term “oxidative phosphorylation” is more appropriate and have made the requested changes.

References 6-9 (line 57) refer to structural studies of the mitochondrial ribosome and not to human diseases and mutations in ribosomal proteins. The authors should use more suitable references.

To put the structures in context with human diseases and mitoribosome assembly, we have added the relevant reference to address this point.

“The central role of mitoribosomes to cellular metabolism is highlighted by several human diseases caused by mutations in either mitoribosomal proteins or assembly factors⁶⁻¹⁰.”

Reference 10:

Lopez Sanchez, M. I. G., Krüger, A., Shiriaev, D. I., Liu, Y. & Rorbach, J. Human Mitoribosome Biogenesis and Its Emerging Links to Disease. *Int J Mol Sci* **22**, 3827 (2021).

Delete hyphen in “mt-SSU” to be consistent (line 195).

This has been corrected.

Referee #2 (Remarks to the Author):

The manuscript by Harper et al. reports the results of a structural investigation by cryo-EM of the assembly of the mitochondrial ribosome (mitoribosome) from yeast and human. It details several intermediate stages of maturation for both the mitochondrial small ribosomal subunits (mtSSU). Among the most interesting findings, the authors describe the involvement of GTPases in the maturation at early stage of the mtSSU decoding center and rRNA processing and folding. Thanks to the parallel investigation of the mtSSU maturation from both yeast and human, the authors are able to unravel several conserved and species-specific maturation steps. Thus, the authors derived several “fundamental principles” for the mtSSU maturation: “1- stepwise activities of molecular switches control early rRNA folding events that lay a foundation for the formation of a functional decoding center”. “2- integration of the rRNA 3’ end facilitates compaction of the functional rRNA core, with yeast and human systems each having evolved distinct solutions to do so”. “3- recognition and processing of 5’ pre-rRNA is a unique feature of the yeast mtSSU assembly pathway, and is carried out by distinct assembly factors”. “4- a conserved assembly factor orchestrates maturation of the head domain and prevents premature engagement of mRNA and the mtLSU”.

In addition to the novel findings, the manuscript details several interesting methodological approaches, such as the use of CRISPR-based biallelic tagging to generate stable HEK293F cells endogenously expressing assembly factor. Such method can be applied for the studies of other factors/proteins involved in a plethora of biological functions.

Finally, the supplementary figures present a wealth of experimental procedures and workflows and will be surely of use for numerous research groups.

The paper is of quality, well written and very clear, the figures are aesthetic and mostly clear (I have few comments on the labeling of the figures), the references appear to be sufficient and the methods are appropriately detailed. One could appreciate the great complementarity this work offers to the recent work from the Amunts lab (Itoh et al., Nature 2022) detailing the maturation of the human mtSSU at a later stage, in addition to the transition from maturation to the translation initiation process.

We thank reviewer 2 for the positive assessment of our work.

As the abundant supplementary information addressed all of my technical questions, I only have minor comments and suggestions to improve the quality of what is already a very smooth manuscript.

The authors write in the abstract “high-resolution structures”. This term is awkward, what is “high-resolution”? better than 3 Å? Up to 2.5 Å? if better than 2 Å, is it still “high-resolution”? It is highly preferable that the authors remove this term and simply state a range of resolution (from 3 to 2.4 Å).

We agree with reviewer 2 and have changed the description of resolution accordingly.

Figure 3, panels A to D, some labels are difficult to read because they are colorful and superimposed to the structure. I suggest to increase the white shadow below or to simply write in black and use arrows instead.

We have revised this figure to address this comment and improve the clarity of the labels.

Figure 4, Panels A to D, similar comment to Figure 3.

We have revised this figure to address this comment and improve the clarity of the labels.

Referee #3 (Remarks to the Author):

The manuscript from Harper et al. is an outstanding and comprehensive study on the maturation of the small ribosomal subunit. The molecular mechanisms described in the manuscript on how the combined action of assembly factors mature the critical functionally regions of the small subunit will transform our understanding of this process. This manuscript provides, in my view, for the first time a comprehensive description of how these regions are assembled in a coordinated manner. I certainly see this manuscript having the impact, novelty and technical quality expected by readers at Nature. I support his publication in Nature after the authors consider my comments below that may add more clarity and relevance to the study.

We thank reviewer 3 for the positive assessment of our work.

The authors determine nine high-resolution structures of in vivo assembled small subunit assembly intermediates from yeast and human mitochondria and use them to describe the maturation process of the decoding center, platform region, processing of the 5' end of the 15S rRNA molecule and head domain. These are extremely complex processes with multiple assembly factors, rRNA helices and r-proteins often acting, changing their conformation, and simultaneously interacting. The authors did an outstanding job in the design of the figures. They significantly assist in understanding the accompanying description of the mechanisms in the text and make the paper easy to follow.

We thank reviewer 3 for the positive feedback on the figures. To comply with editorial guidelines, we have combined figures 1 and 2, which we believe is the only way we can meet the limit of six figures while maintaining the clarity of the description in the text.

A strength of the study is the comparative done for the maturation process between human mitochondria and yeast. I found particularly intriguing the description of the maturation mechanism for the decoding center. GTPase NOA1, methyltransferase TFB1M (homologue to bacterial KsgA) and RBFA control the maturation of the decoding center. This includes the docking of helix 44, mainly regulated by NOA1, and positioning of helices 44, 27 and 45 after NOA1 and TFB1M leave to create the binding site for mS38. While I agree with the interpretation of the structures, the chronologic description of events described in the manuscript is presented as the only possible sequence of events. Similarly, the maturation process described here for the head, platform and processing of the 5' end, all processes are described as occurring in a strictly linear fashion.

Whereas the cryo-EM structures likely describe the most frequent or 'canonical' maturation pathway, it is well established that ribosome biogenesis occurs through multiple parallel assembly pathways. That concept is completely absent from the mechanistic descriptions in this manuscript. All is described in a strict chronology of events that occur in a linear fashion. I believe this concept should be integrated across the entire manuscript and in interpreting all the maturation events described for each region.

We thank reviewer 3 for this comment and have addressed parallel assembly pathways together with the subsequent point (see below).

A second point I would like the authors to clarify is the criteria used for the chronological arrangement of the structures obtained for the assembly intermediates. I see the absence and presence of assembly factors, as well as the observed protein complement in the intermediates, as the data that guide the authors in the placement of the intermediates in the linear timeline. This interpretation of the structures assumes a somehow 'simplistic' view of how assembly factors and r-proteins promote assembly. It assumes factors and ribosomal proteins bind once, act and either leave (factors) or stay (r-proteins). Is it possible that some of the factors may need to bind and be released multiple times to catalyze the folding of a particular site? Is it possible a factor may act at different times

in the linear timeline and catalyze various steps on the folding of a region? Is it possible these two events catalyzed by the same factor may occur at different times because they are done through functional interplays with other assembly factors?

We thank reviewer 3 for raising this point as it allows us to highlight that on page 5, we have already indicated that the “maturation status of rRNA elements” as well as “the presence of assembly factors” were the main factors leading to the assignment of different assembly intermediates. We further agree with reviewer 3 that the presence of parallel assembly pathways needs to be clarified more directly. The relevant paragraph on page 5 now reads:

“These intermediates can be ordered into an assembly pathway that rationalizes the maturation status of rRNA elements as well as the presence of assembly factors and mitochondrial proteins. While our assignment of states A-E suggest a linear assembly pathway of intermediates bound by METTL17, we cannot exclude the possibility of alternative pathways present in the cell, including those in which certain proteins undergo multiple cycles of binding and dissociation. In particular, the identification of the less populated assembly state C* suggests uncoupling of head and body assembly mechanisms during maturation, supporting the notion that mtSSU assembly does not occur in a strictly linear fashion, in line with the parallel assembly pathways previously characterized for bacterial SSU assembly²⁵ (Supplementary Figs. 9, 18).”

Reference 25:

Sykes, M. T. & Williamson, J. R. A complex assembly landscape for the 30S ribosomal subunit. *Annu Rev Biophys* **38**, 197–215 (2009).

It is likely that if any of these types of events are occurring (parallel pathways of assembly and factors acting multiple times), the nine presented structures may not be sufficient to provide a complete description of the maturation events. This is undoubtedly a limitation of this study that should be acknowledged by the authors as part of the discussions. As of written, the manuscript describes the maturation process of the small ribosomal subunit as a process occurring in a strict sequence of events. This model does not align well with our current understanding of the ribosome assembly process as a flexible and plastic process capable to occurs through multiple pathways in which maturation steps do not necessarily occur in a strict order.

We agree with reviewer 3 that the provided structural insights presented in this study are not yet exhaustive but rather provide all assembly intermediates that can be obtained using a single protein bait, in this instance METTL17/Rsm22. As such, further mechanistic studies will be required in the future to reveal the degree to which parallel pathways can be applied to mitochondrial small subunit assembly under native conditions. To address this more specifically in the text in the discussion, we have added the following sentence on page 17:

“On the contrary, the shared architecture and role of Rsm22/METTL17 in prevention of mRNA binding, subunit compaction, and decoding center formation demonstrates a level of conservation between these two assembly systems, a concept additionally highlighted by similarities in rRNA folding chronology (Figs. 6, 1i,j). We further note that our study provides a structural view of all assembly intermediates that can be associated with Rsm22/METTL17, thereby visualizing one assembly line of what is likely a branched assembly landscape involving assembly factors dissociating and re-associating at key junctions.”

Reviewer Reports on the First Revision:

Referees' comments:

Referee #1 (Remarks to the Author):

Point 1:

There are no doubts that the manuscript by Harper et al is a great study suitable for Nature. However, although the authors have addressed my comments, I am disappointed that they do not provide experimental proof that GFP tagged METTL17 is functional. This could have been easily done without much effort (e.g. even western blot analysis of mtDNA-encoded proteins like COX1 would have been sufficient to show that mitoribosome function is not impaired).

Point 2:

I cannot follow this argumentation. The ribosome complexes were isolated via METTL17, which is not part of the mature mtSSU.

Here, I only wanted to make the point that it would be appreciated by the reader to put the current findings into the context of the study by Ito et al. (2022, Nature).

Referee #3 (Remarks to the Author):

I appreciate the authors considering my comments. I think this new version of the manuscript cleared all my concerns, and I am supportive of publication in Nature as is.

Author Rebuttals to First Revision:

Response to the reviewers

Referees' comments:

Referee #1 (Remarks to the Author):

Point 1:

There are no doubts that the manuscript by Harper et al is a great study suitable for Nature. However, although the authors have addressed my comments, I am disappointed that they do not provide experimental proof that GFP tagged METTL17 is functional. This could have been easily done without much effort (e.g. even western blot analysis of mtDNA-encoded proteins like COX1 would have been sufficient to show that mitoribosome function is not impaired).

We thank reviewer 1 for this clarification and have performed Western blots to show that the expression of mitochondrially encoded protein genes (here COX1 and ND1) is unaffected by the tagging of METTL17 (Supplementary Figures 1b, 2f). We believe that these data show that mitoribosome function is not impaired and in the main text this is further referenced in lines 97-98, where we state:

“...and confirmed that tagging does not affect mitochondrial translation (Supplementary Fig. 2)²⁴.”

Point 2:

I cannot follow this argumentation. The ribosome complexes were isolated via METTL17, which is not part of the mature mtSSU.

Here, I only wanted to make the point that it would be appreciated by the reader to put the current findings into the context of the study by Ito et al. (2022, Nature).

We thank reviewer 1 for clarifying this point. We agree that contextualizing our data in light of the study by Itoh et al. is important and have addressed this in the following statements in the text:

- 1. For overall clarity, in the Discussion, we state that our human mtSSU assembly intermediates are located upstream of those observed in Itoh et al.**

“In the human assembly system, we show that assembly factors in concert with mitoribosomal proteins catalyze stepwise formation and stabilization of rRNA elements prior to states previously observed²¹.” (lines 377-379)

- 2. More specifically, we address the context in which METTL17 acts with direct comparison of METTL15 as outlined in lines 319-326:**

“The binding of METTL17 to h31 also prevents association of the late-stage assembly factor METTL15, which is only observed in assembly intermediates with a mature head domain²¹. These events enforce a chronology of human mtSSU assembly in which head formation and compaction precedes final

METTL15-mediated modification of the decoding center. The architectural role of METTL17 and position in the assembly pathway upstream of METTL15 would further explain rRNA methylation defects observed in METTL17-deficient cells⁴⁷.

3. To further contextualize the position of our assembly intermediates, we highlight the long residence time of RBFA, which continues to be part of later assembly intermediates as described by Itoh et al. (lines 240-242)

“Maturation events downstream of State E subsequently link RBFA and late-stage assembly factor METTL15 function during mitoribosome assembly with mtIF3-mediated translation initiation²¹.”

As listed in the comments above, we believe that our assembly intermediates occur before those observed by Itoh et al. based on the conformation of rRNA and presence/absence of proteins in both sets of structures. Beyond a strictly linear pathway (as noted by reviewer 3), cycles of binding and dissociation of assembly factors will occur to give rise to what is likely a branched assembly pathway, with isolated complexes representing different portions of the pathway.

Referee #3 (Remarks to the Author):

I appreciate the authors considering my comments. I think this new version of the manuscript cleared all my concerns, and I am supportive of publication in Nature as is.

We thank reviewer 3 for the positive assessment of the revised manuscript.

Reviewer Reports on the Second Revision:

Referees' comments:

Referee #1 (Remarks to the Author):

I thank the authors for addressing all my comments. I highly appreciate the experimental proof for the function of GFP-tagged METTL17 and the extended discussion. The included contextualization with the data by Itoh et al. increases the added value of this anyway high-quality study. I strongly support the manuscript to be published in Nature.